# Learning with Holographic Reduced Representations

**Ashwinkumar Ganesan[1]**\***Hang Gao[1], Sunil Gandhi[1]**\***Edward Raff[1,2,3], Tim Oates[1]**
**James Holt[2], Mark McLean[2]**
[1]University of Maryland, Baltimore County
[2]Laboratory for Physical Sciences
[3]Booz Allen Hamilton

## Abstract

Holographic Reduced Representations (HRR) are a method for performing symbolic AI on top of real-valued vectors [1] by associating each vector with an abstract concept, and providing mathematical operations to manipulate vectors as if they were classic symbolic objects. This method has seen little use outside of older symbolic AI work and cognitive science. Our goal is to revisit this approach to understand if it is viable for enabling a hybrid neural-symbolic approach to learning as a differentiable component of a deep learning architecture. HRRs today are not effective in a differentiable solution due to numerical instability, a problem we solve by introducing a projection step that forces the vectors to exist in a well behaved point in space. In doing so we improve the concept retrieval efficacy of HRRs by over $100\times$. Using multi-label classification we demonstrate how to leverage the symbolic HRR properties to develop an output layer and loss function that is able to learn effectively, and allows us to investigate some of the pros and cons of an HRR neuro-symbolic learning approach. Our code can be found at https://github.com/NeuromorphicComputationResearchProgram/Learning-with-Holographic-Reduced-Representations

## 1 Introduction

Symbolic and connectionist (or "neural") based approaches to Artificial Intelligence (AI) and Machine Learning (ML) have often been treated as two separate, independent methods of approaching AI/ML. This does not need to be the case, and our paper proposes to study the viability of a hybrid approach to propagation based learning. In particular, we make use of the Holographic Reduced Representation (HRR) approach originally proposed by Plate [1]. Plate proposed using circular convolution as a "binding" operator. Given two vectors in a $d$ dimensional feature space, $\boldsymbol{x}, \boldsymbol{y} \in \mathbb{R}^d$, they can be "bound" together using circular convolution, which we denote as $\boldsymbol{s} = \boldsymbol{x} \otimes \boldsymbol{y}$. This gives us a new result $\boldsymbol{s} \in \mathbb{R}^d$. The HRR approach also includes an inversion operation † that maps $\mathbb{R}^d \to \mathbb{R}^d$.

With this binding and inverse operation, Plate showed that we can assign vectors to have a conceptual symbolic meaning and construct *statements* (or "sentences") from these representations. For example, we can construct $S = red \otimes cat + blue \otimes dog$ to represent "a red cat and a blue dog". HRRs can also *query* the statement representation $S$. To ask which animal was red, we compose $S \otimes red^\dagger \approx cat$ which gives us a numeric output approximately equal to the vector representing "cat".

The HRR framework requires a constant amount of memory, relies on well-optimized and scalable operations like the Fast Fourier Transform (FFT), provides symbolic manipulation, and uses only differentiable operations, that would make it seem ideal as a tool for exploring neuro-symbolic modeling. Unfortunately back-propagating through HRR operations does not learn in practice,

---

\*Research work completed prior to joining Amazon.

35th Conference on Neural Information Processing Systems (NeurIPS 2021).

rendering it seemingly moot for such neuro-symbolic research. The goal of this work is to find sufficient conditions for successful back-propagation based learning with the HRR framework, and to develop evidence of its potential for designing future nero-symbolic architectures. Because the utility of HRRs for cognitive-science tasks are already well established, our application area will focus on a multi-label machine learning task to show how HRRs can provide potential value to research in areas it has not been previously associated with.

Backpropagating through HRRs is ineffective if done naively, and our goal is to show how to rectify this issue with a simple projection step and build viable loss functions out of HRR operations. We choose to do this with eXtreme Multi-Label (XML) classification tasks as one that is impossible for HRRs to tackle today. XML tasks have an input $\boldsymbol{x}$, from which we have a large number $L$ of binary prediction problems $\mathcal{Y}_1, \mathcal{Y}_2, \ldots, \mathcal{Y}_L$. Our approach will be to represent each of the $L$ classes as a HRR concept vector, and construct an alternative to a fully connected output layer that uses the HRR framework for learning. Our experiments and ablations will focus on the impact of replacing a simple output layer with HRRs and other changes to the HRR approach to better understand its behaviors. State-of-the-art XML performance is not a goal.

Our contributions and the rest of our paper are organized as follows. In §2 we discuss prior approaches related to our own, and how our work differs from these methods. In §3 we give a brief overview of the HRR operator's details given its niche recognition within the AI/ML community, and introduce a simple improvement that increases the binding stability and effectiveness by $100\times$. Then in §4 we show how we can leverage HRR's symbolic nature to create a new output layer and loss function combination that provides several benefits to deep extreme multi-label models. With this new method we demonstrate in §5 that compared to a simple fully-connected output layer we can reduce the output size by as much as $99.55\%$, resulting in a total reduction in model size by up to $42.09\%$, reduce training time by $41.86\%$, and obtain similar or improved relative accuracy by up to $30\%$. We also show that these results are often competitive with more advanced approaches for XML classification, though do still suffer at the largest output spaces with $670k$ labels. Finally we conclude in §6.

## 2 Related Work

Smolensky [2] elucidated early arguments for neuro-symbolic approaches with the Tensor Product Representation (TPR) that defined the first Vector Symbolic Architecture (VSA), defining binding and unbinding operations that allow symbolic style object manipulation atop some vector field. As argued by Smolensky [2] (and echoed by Greff et al. [3]), symbolic logic has greater power for complex tasks, but is brittle in requiring precise specification, where connectionist deep learning is more robustly able to process the raw inputs to a system, its capacity for more powerful "logic" is still limited and hotly debated. Combining these two domains in a neuro-symbolic hybrid system may yield positive results, and several recent works have found success in natural language processing tasks by augmenting recurrent neural networks with TPR inspired designs [4–6]. However, binding $n$ vectors that are each $d$ dimensions requires $\mathcal{O}(d^n)$ space using TPRs. Thus VSAs that require a fixed dimension are computational preferable, but no work to our knowledge has successfully used a VSA of fixed length in gradient based learning. Schlegel et al. [7] provide a comprehensive summary and comparison of fixed length VSAs, and the details of how we chose to select HRRs based on their potential for gradient based learning and compute/memory efficacy in current frameworks like PyTorch are presented in Appendix D. Though such fixed-length VSAs have not been used in gradient based learning, they have still proven effective low-power embedded computation enviroments due to low-level hardware optimizations possible with many VSAs [8, 9].

The HRR operation has found significant use in cognitive science to create biologically plausible models of human cognition [10–13]. These approaches use the HRR's operations in a primarily symbolic fashion to demonstrate human-like performance on a number of cognitive science tasks [14]. The vector representation is used as the foundation for biological plausibility after establishing that the operations required to implement HRR are within the plausible sphere of a biological substrate [15]. When learning has been incorporated in these models it has been primarily through examples of Hebbian updates of spiking models on the outputs of HRR, rather than back-propagation through the HRR operations and preceding inputs/layers [16]. In our work we advance the understanding of how to learn through the HRR through propagation in an effective manner and how HRR can be leveraged as an integral part of a neural-symbolic approach.

Little work exists on gradient based learning with HRR operations explicitly. The only work we are aware that attempts to leverage the symbolic properties of the HRR is by Nickel et al. [17], who use the binding operations to connect elements in a knowledge graph as a kind of embedding that combines two vectors of information without expanding the dimension of the representation (e.g., concatenation would combine but double the dimension). More recently Liao and Yuan [18] used circular convolution as a direct replacement of normal convolution to reduce model size and inference time, without any leverage of the symbolic properties available within HRRs. While Danihelka et al. [19] purport to embed the HRR into an LSTM, their approach only augments an LSTM by including complex weights and activations[2], and does not actually use HRR as it lacks circular convolution, and all three works do not leverage the inverse operation † or seek to leverage any symbolic manipulations. By contrast our work explicitly requires the symbolic manipulation properties of HRR to create a vector-symbolic loss function, and we need to leverage the inverse operator †, meaning we use the entire HRR framework. We also demonstrate new insights into learning with HRR operations that make them more effective, where prior works have simply used the $\otimes$ operator on $\leq 2$ components without further study.

There has been other work in differentiable neuro-symbolic systems outside of TPRs and VSAs, most notably using first-order logic [20–22]. These works represent another powerful alternative approach, though at this time are often more involved in their design and training procedures. We consider these beyond the scope of our current study, where we seek to obtain similar benefits using the simpler HRR that requires only the careful application of FFTs to leverage. This is an easily satisfiable constraint given their heavy optimization and wide spread use amongst existing deep learning frameworks.

Our use of extreme multi-label classification problems is due to it being out-of-reach of current HRR methods, which we found produced random-guessing performance in all cases. There exists a rich literature of XML methods that tackle the large output space from the perspective of decision trees/ensembles [23–27], label embedding regression [28–31], naive bayes [32], and linear classifiers [33, 34]. There also exist deep learning XML methods that use either a fully-connected output layer [35] and others that use a variety of alternative approaches to dealing with the large output space [36–43]. The variety of approaches is orthogonal to the purpose of this work, which is to investigate HRRs. We will leverage only some of these works to modify their architectures to use a fully-connected output layer (if they do not already) and to use our HRR approach, so that we can observe the impact of the HRR specifically on the model. In our experimentation we found that the mechanisms for handling the high cardinality of outputs is highly architecture tuned and specific, making it difficult to do broader comparisons. Our work will explore fully-connected, convolutional, and recurrent architectures to show that our HRR approach can work broadly across many types of architectures that have been used, but it is not our goal to be the "best" at XML by any metric. The XML task's value to our study is a problem that current HRR methods could not tackle, and where we can apply the symbolic properties of HRR in developing the solution in a way that demonstrates how one may design HRR based networks for other tasks.

## 3   Holographic Reduced Representation

The original paper by Plate [1] introduced the HRR approach and developed significant theory for its use as a symbolic approach to AI on a neural substrate. We will quickly review its foundation and numerical issues that lead to our projection based improvement. The HRR starts with circular convolution as a "binding" operator. We use Plate's notation of $\otimes$, and a definition of circular convolution using an FFT $\mathcal{F}$ and inverse FFT ($\mathcal{F}^{-1}$) is given in eq. (1). It is easy to see that the binding operator of circular convolution $\otimes$ is commutative. What is particularly interesting is the ability to *unbind* terms that have been bound by this operator. Let $\mathcal{F}(\boldsymbol{a})_i$ denote the $i$'th complex feature of the FFT of $\boldsymbol{a}$. By defining an identity function $\mathcal{F}(\boldsymbol{a}^\dagger)_i \mathcal{F}(\boldsymbol{a})_i = 1$, we can derive the inverse function eq. (2), where we raise each complex coefficient $z_i$ of the FFT to the $z_i^{-1}$ power before going back to the real domain.

$$\boldsymbol{a} \otimes \boldsymbol{b} = \mathcal{F}^{-1}\left(\mathcal{F}(\mathbf{a}) \odot \mathcal{F}(\mathbf{b})\right) \qquad (1)$$

$$\boldsymbol{a}^\dagger = \mathcal{F}^{-1}\left(\frac{1}{\mathcal{F}(\boldsymbol{a})}\right) \qquad (2)$$

---

[2]HRRs & circular convolution can naturally exist in the reals.

Using this, Plate [1] showed that one can compose sentences like our example $S = red \otimes cat + blue \otimes dog$ by assigning concepts to vectors in a $d$ dimensional space. We can unbind components with the inverse operation, giving $S \otimes cat^\dagger \approx red$, or we can check for the existence of a term by checking that $S^\top cat^\dagger \approx 1$. A term not present, like "cow", would behave as $S^\top cow^\dagger \approx 0$. These sentences can be constructed with more complex logic. The binding is also distributive (and associative), so $\boldsymbol{a} \otimes (\boldsymbol{x} + \boldsymbol{y}) = \boldsymbol{a} \otimes \boldsymbol{x} + \boldsymbol{a} \otimes \boldsymbol{y}$. This allows us to construct complex symbolic relationships. As an example, we could have: $S = first \otimes (name \otimes Jane + age \otimes 30) + second \otimes (name \otimes Joe + age \otimes 33)$, and to determine the name of the first person we can perform $first^\dagger \otimes name^\dagger \otimes S \approx Jane$.

To make this symbolic manipulations effective, Plate [1] introduced two points of sufficiency. First, a vector $\boldsymbol{a}$ used for HRR operations should have its values sampled from a normal distribution: $\boldsymbol{a} \sim \mathcal{N}\left(0, I_d \cdot d^{-1}\right)$. This allows binding and unbinding to work in expectation. Second, is that the inverse $\dagger$ is numerically unstable. The remedy developed is the *pseudo* inverse $*$, given by $\boldsymbol{a}^* = [a_1, a_d, a_{d-1}, \ldots, a_2]$. This approximation was justified by noting that in polar form one obtains $\mathcal{F}_j\left(\mathbf{a}^\dagger\right) = \frac{1}{r_j} e^{-i\theta_j}$ and $\mathcal{F}_j\left(\mathbf{a}^*\right) = r_j e^{-i\theta_j}$, which are off by the reciprocal of the complex magnitude. Plate and others [14] use the pseudo inverse $*$ exclusively as the error due to approximation of $\dagger$ was smaller than the errors caused by numerical instability. In Appendix B we provide additional illustration about the potential modeling with HRRs, and worked examples demonstrating why the math works is provided in Appendix B.

### 3.1 Improved HRR Learning

The only prior work backpropagating through $\otimes$ only needed to bind two items together at a time [17], where we need to bind and represent tens-to-hundreds of thousands of concepts. Prior work also made no use of the unbinding with $\dagger$, which we require. We found that the default method proposed by Plate [1] was not sufficient for our work, and needed improvement. In particular, we needed better detection if a vector $\boldsymbol{y}$ (read, *label*) has been bound to a statement $S$ (read, *output*). We achieve this by defining a new *complex unit magnitude projection*, which is given by eq. (3).

We change the initialization to $\boldsymbol{a} \sim \pi\left(\mathcal{N}\left(0, I_d \cdot d^{-1}\right)\right)$ which ensures the vectors we use for HRR operations are unitary, meaning the complex magnitude is one. In this case we obtain $\boldsymbol{a}^\dagger = \boldsymbol{a}^*$ because in polar form we have $r_j = 1 \; \forall j$, giving $\mathcal{F}_j\left(\mathbf{a}^\dagger\right) = \frac{1}{1} e^{-i\theta_j} = \mathcal{F}_j\left(\mathbf{a}^*\right) = 1 e^{-i\theta_j}$.

$$\pi(\boldsymbol{x}) = \mathcal{F}^{-1}\left(\ldots, \frac{\mathcal{F}(\boldsymbol{x})_j}{|\mathcal{F}(\boldsymbol{x})_j|}, \ldots\right) \tag{3}$$

This has a number of benefits. First, we make the numerically stable $*$ have a mathematically equivalent result to the true inverse, removing that source of error. Second, the $*$ is faster to calculate requiring a simple shift of values that takes $\mathcal{O}(d)$ time to perform, compared to $\mathcal{O}(d \log d)$ for the FFTs in $\dagger$. Third, we find that this complex projection significantly increases the number of concepts that can be bound together and the accuracy of retrieving them. Using standard HRR initialization we found it difficult to use more than 10 bindings in a statement, but our simple change allows us to improve query accuracy by up to $100\times$ as many bound vectors.

Figure 1: Demonstration of the HRR query checking if $\boldsymbol{x} \otimes \boldsymbol{y}$ is in the HRR vector $S$. If present, the response value (y-axis) should return $\approx 1.0$ (dashed line). If absent, the response should $\approx 0$ (dotted line). Naive use of HRR (blue & green) results in more noise with present and absent results both returning values far above and below the expected range $[0, 1]$. Our projection $\pi$ (orange & red) keeps values stable and near their expected values.

To demonstrate the effectiveness of our projection step, we create a statement vector $S = \sum_{i=1}^{T} \boldsymbol{a}_i \otimes \boldsymbol{b}_i$. We can ask if two vectors $\boldsymbol{x} \otimes \boldsymbol{y} \in S$ by checking that $\boldsymbol{x}^\top (S \otimes \boldsymbol{y}^\dagger) \approx 1$. If the pair are not bound in $S$, then $\boldsymbol{x}^\top (S \otimes \boldsymbol{y}^\dagger) \approx 0$ meaning $\boldsymbol{x} \otimes \boldsymbol{y} \notin S$. We explicitly use this problem formulation because it is integral to our approach to XML. This is shown for a 256 dimension space in Fig. 1 where the black dashed

line marks the "in" case and the dotted black line the "not in" case. Notice that the naive HRR approach has present values that look absent, and absent values that look present, with unexpectedly large and small magnitudes. The variance of naive HRR also increases as the number of terms bound together increases. In contrast, our projection $\pi$ keeps the response around 1 and 0 for present and absent tokens respectively, even when binding 1024 vectors in only 256 dimensions. This is critical for us to represent large output spaces in a smaller source dimension, and removes the need for complex averaging and associative cleanup approaches attempt by Danihelka et al. [19]. Plate's original theory indicated binding capacity would increase linearly with dimension size $d$, which we found did not hold for naive HRRs, but does for our projected variant. Additional details and experiments on binding capacity, and selection of HRRs as opposed to alternative binding operations, is provided in Appendix D.

## 4   Dense Label Representations

Now we will define our new neuro-symbolic approach to XML tasks by leveraging the HRR operations with our improved initialization from §3.1. We define $L$ as the number of symbols (or labels) in the given task dataset. We will use a $d' \ll L$ dimensional output space $\mathbb{R}^{d'}$, which is a hyper-parameter we can choose. We will define two vectors $\boldsymbol{p}, \boldsymbol{m} \in \mathbb{R}^d$ to represent the concepts of a class being *present* and *missing* (or "negative" / "absent") from the label space $\mathcal{Y}_{1,\ldots,L}$. We initialize $\boldsymbol{p} \sim \pi(\mathcal{N}(0, \mathrm{I}_d \cdot d'^{-1}))$, and then select $m$ to be any vector orthogonal to $\boldsymbol{p}$.

We now define some additional notation and terms to describe our approach. We will use $\mathcal{Y}^p$ to denote the set of ground truth labels present for a given datapoint $\boldsymbol{x}$ and $\mathcal{Y}^m$ for the missing labels, where $\mathcal{Y}^p \cup \mathcal{Y}^m = \mathcal{Y}_{1,2,\ldots,L}$. For every individual classification problem $\mathcal{Y}_i$, we will use $\boldsymbol{c}_i \sim \pi(\mathcal{N}(0, \mathrm{I}_d \cdot d'^{-1}))$ to represent each class with a vector. The vectors $\boldsymbol{p}, \boldsymbol{m}, \boldsymbol{c}_{1,2,\ldots,L}$ will all be initialized as we have described, and will not be altered during training. To denote the neural network we will train and its output, we will use $f(\boldsymbol{x}) = \hat{\boldsymbol{s}} \in \mathbb{R}^{d'}$.

**HRR Label Vector:** We begin by converting the labels for a data point $\boldsymbol{x}$ into a *labels vector* $\boldsymbol{s} \in \mathbb{R}^{d'}$. This vector $\boldsymbol{s}$ will be constructed using the HRR framework to be a neuro-symbolic representation of the original labels, and is defined by eq. (4). Thanks to the the commutative property of $\otimes$, the "present" vector $\boldsymbol{p}$ can move outside the summation and be bound with all present classes, and the non-present vector $\boldsymbol{m}$ can be bound with every item not present. This will create a statement $\boldsymbol{s}$ with technically hundreds of thousands of bound components that we could query. We can compute this efficiently in $\mathcal{O}(|\mathcal{Y}^p|)$ by computing once the "all labels" vector $\mathcal{A} = \sum_{i=1}^{L} \boldsymbol{c}_i$. This is done once at initialization, and then we can re-write eq. (4) into the more efficient form of eq. (5), leveraging the symbolic properties of HRR to avoid computation.

$$\boldsymbol{s} = \overbrace{\sum_{\boldsymbol{c}_p \in \mathcal{Y}^p} \mathbf{p} \otimes \boldsymbol{c}_p}^{\text{Labels present}} + \overbrace{\sum_{\boldsymbol{c}_m \in \mathcal{Y}^m} \boldsymbol{m} \otimes \boldsymbol{c}_m}^{\text{Labels absent}} \quad (4) \qquad = \mathbf{p} \otimes \sum_{\boldsymbol{c}_p \in \mathcal{Y}^p} \boldsymbol{c}_p + \boldsymbol{m} \otimes \left( \mathcal{A} - \sum_{\boldsymbol{c}_p \in \mathcal{Y}^p} \boldsymbol{c}_p \right) \quad (5)$$

**HRR XML Loss:** We now have the output of our network $\hat{\boldsymbol{s}}$ and a target vector $\boldsymbol{s}$ that exist in a $d' \ll L$ dimensional space. A straightforward idea would be to take an embedding style approach to make the loss function $\ell(\boldsymbol{s}, \hat{\boldsymbol{s}}) = \|\boldsymbol{s} - \hat{\boldsymbol{s}}\|_2^2$. This is intuitive and matches prior approaches to XML [28, 30]. However, we found that such an approach resulted in no learning and degenerate random-guessing performance. This is because the regression formulation targets the overall errors in exact values, rather than the real goal of being able to extract the present classes from $\hat{\boldsymbol{s}}$. Because binding many values in a fixed dimension $d'$ introduces noise, the regression approach attempts to learn this underlying noise that is not meaningful to the actual problem.

Our solution to this issue is to again leverage the properties of HRR operations to create a loss based on our ability to accurately query the predicted statement $\boldsymbol{s}$ for the same components we would expect to find in the true label $\boldsymbol{s}$. We remind the reader that we can query if a class $\boldsymbol{c}_p \in \boldsymbol{s}$ by checking if $\boldsymbol{s}^\top \boldsymbol{c}_p^* \approx 1$. We normalize this dot product, i.e., use the cosine similarity $\cos(\boldsymbol{s}, \boldsymbol{c}_p^*) = \frac{\boldsymbol{s}^\top \mathbf{c}_p^*}{\|\boldsymbol{s}\|\|\mathbf{c}_p^*\|}$ to perform these queries in a manner that limits the response magnitude to $[-1, 1]$, which will prevent degenerate solutions that attempt to maximize magnitude of response over query efficacy.

With the bounded cosine response, our loss has two components. First we build a loss term that goes though all present labels $\mathcal{Y}^p$, checking that each can be extracted. This involves computing $\mathbf{p}^* \otimes S$ to extract the symbolic statement representing all present vectors (i.e., $\mathbf{p}^* \otimes S \approx \sum_{\boldsymbol{c}_p \in \mathcal{Y}^p} \boldsymbol{c}_p$). Then we use each label HRR vector $\boldsymbol{c}_p$ and check if it is present in the extracted output. This results in eq. (6)

Next we confirm that the absent labels (the vast majority) are *not* queryable from the output. Rather than enumerate all $L - |\mathcal{Y}^p|$ labels, or perform lossy negative sampling, we can instead leverage the symbolic properties of HRR. We will compute $\boldsymbol{m}^* \otimes \boldsymbol{s}$ to extract the representation of all non-present classes, and perform a dot product against all present classes that are expected. This gives us eq. (7).

$$J_p = \sum_{\boldsymbol{c}_p \in \mathcal{Y}^p} \left(1 - \cos\left(\mathbf{p}^* \otimes \hat{\mathbf{s}}, \boldsymbol{c}_p\right)\right) \quad (6) \qquad J_n = \cos\left(\mathbf{m}^* \otimes \hat{\mathbf{s}}, \sum_{\boldsymbol{c}_p \in \mathcal{Y}^p} \boldsymbol{c}_p\right) \quad (7)$$

This negative component of the loss works because it is minimized by having the absent labels $\boldsymbol{m}^* \otimes \boldsymbol{s}$ and present labels $\sum_{\boldsymbol{c}_p \in \mathcal{Y}^p} \boldsymbol{c}_p$ be disjoint (i.e., no overlap). If there is any overlap, the dot product between these terms will increase, and thus, increase the error.

Through this neuro-symbolic manipulation we can create a loss term that simultaneously considers the presence/absence of all $L$ labels in only $d' \ll L$ dimensions and $\mathcal{O}(|\mathcal{Y}^p|)$ time. The final loss is simply $\ell(\boldsymbol{s}, \hat{\boldsymbol{s}}) = J_p + J_n$.

We remind the reader that the target label $\boldsymbol{s}$ is present in the form of knowing the exact $\boldsymbol{c}_p \in \mathcal{Y}^p$ vectors to use. One could, if preferred, denote this loss as $\ell(\mathcal{Y}^p, \hat{\boldsymbol{s}})$. We also note that expectation may be that a new hyper-parameter $\lambda$ is needed to balance between the relative importance of eq. (6) and eq. (7). However, we find that no such parameter is needed and simply adding them together is effective.

## 5   Experiments & Analysis

We will now conduct an experimental evaluation of our new HRR based approach to XML classification. We emphasize to the reader that this is to show that the HRR approach of symbolically modeling a goal, and then back-propagating through the vector-symbolic HRR, works and has potential utility broadly. We do not argue for "dominance" at the XML task or to supplant all current approaches. XML is simply a task we are applying the HRR to show the pros (and some cons) of the HRR approach. To this end we will show results with a Fully Connected (FC), Convolutional Neural Network (CNN) based network, and attention augmented Long-Shot Term Memory (LSTM) networks. The CNN and LSTM models are applied to every dataset for which the original text was available, only FC is applied to every dataset since fixed-length feature vectors were always provided[3]. Each will be endowed with our HRR dense label representations as described in §4 using our improved HRR initialization from §3 (becoming HRR-FC, HRR-CNN, and HRR-LSTM). The CNN & LSTM are taken from the XML-CNN [35] and AttentionXML [41] works. The AttentionXML's hierarchical prediction head is replaced with a fully connected layer in our experiments because we are interested in studying the impact of FC → HRR replacement, and how the change to an HRR impacts the standard network behavior. Using the XML tasks we will study the impact of the HRR an ablate choices like the use of fixed vs learned values for the $\boldsymbol{p}$ & $\boldsymbol{m}$ vectors, as well as the label vectors $\boldsymbol{c}_p$, impact of increasing the HRR dimension $d'$, and impact on model size and training time. We also ablated the impact of our projection step $\pi(\cdot)$, and found that in *all* cases skipping the projection step caused degradation to random-guessing performance, and thus will not be explicitly included in results.

### 5.1   Datasets & Evaluation Metrics

To assess the impact of dense label representations, we use eight of the datasets from Bhatia et al. [44]. Table 6 (see appendix) provides statistics about each dataset used for experiments. The average number of samples per label varies from 2.29-1902.15, and the average number of labels per data point varies from 2.40-75.54. Bhatia et al. [44] split the dataset into small scale and large datasets depending on the number of labels in each input sample. Small scale datasets consist of at most 5000

---

[3]We did contact [44] for all text data but received no reply.

labels. The features are a bag-of-words representation of the text in every dataset. Negative labels are as much as $124321\times$ more populous than positive labels per point. TF-IDF style pre-processed features are available for all datasets, but the original text is not. We were unable to obtain the original text for all datasets, and models which require the original text (CNN & LSTM) thus are not tested when the raw text is not available.

We will consider two primary metrics in our evaluation that are common in XML work, Precision at $k$ (P@$k$) and the propensity score at $k$ (PSP@$k$). Given $\text{rank}_k(\hat{\mathbf{y}})$ as the rank of all the labels in $\hat{\mathbf{y}}$ and $p_l$ is the relative frequency of the $l$'th label, the P@$k$ (Equation 8) measures raw predictive accuracy of the top-$k$ guesses, and PSP@$k$ (Equation 9) down-weights correctly predicting frequent classes to counteract the power-law distribution of labels common in XML problems.

$$\text{P@}k := \frac{1}{k} \sum_{l \in \text{rank}_k(\hat{\mathbf{y}})} \mathbf{y}_l \qquad (8) \qquad\qquad \text{PSP@}k := \frac{1}{k} \sum_{l \in \text{rank}_k(\hat{\mathbf{y}})} \frac{\mathbf{y}_l}{p_l} \qquad (9)$$

For brevity we will focus on $k = 1$ in most experiments, but found that $k \in [1, 5]$ scores were all highly correlated and did not meaningfully alter any results. Additional metrics we considered are mentioned in Appendix F, but we found them so highly correlated with either P@$k$ or PSP@$k$ as to make them redundant.

## 5.2  Network Architectures

The baseline multi-label network is a fully-connected (FC) network with two hidden layers. Both hidden layers have the same size of $512$ with a ReLU activation. The basline network has $L$ outputs trained with binary cross entropy (BCE) loss with appropriate sigmoid activation. Our HRR version of this network (HRR-FC) uses the same architecture for input and hidden layers. But unlike the multi-label network, the output layer is significantly constrained (size is $512\times d'$) where $d'$ is the size of dense label representation (§4 for more details). We note that this gives our HRR Network less parameters to solve the same problem, giving it a disadvantage in capacity.

For a CNN based model we will use the XML-CNN approach of [35]. Their original architecture with their code is used as the baseline, and our modified version (HRR-CNN) has the same architecture but replaces the output layer with the HRR approach as we used in HRR-FC just described. We note that the original code selected best results from the test set, leading to over-fitting. We have corrected this issue which prevents us from achieving the same results previously published.

For the LSTM model, we use the attention based bidirectional approach of AttentionXML [41]. The approach used for the output prediction of AttentionXML is involved in a hierarchical sequence of fully connected layers applied per output token to produce a single scalar output, which makes it non-viable to directly convert to an HRR based approach. For this reason we replace the final hierarchical prediction heads of AttentionXML with the same attention mechanism but use a standard fully-connected output layer like the FC and XML-CNN models do, and denote this as the "LSTM" model in results. Our version of AttentionXML with the dense HRR label approach is denoted as HRR-LSTM.

Our goal is not to determine the most accurate possible XML model, and for this reason we do not perform any expensive hyper-parameter search over the architectures for any of the six models considered (FC, CNN, LSTM, and HRR variants). Doing so to maximize model accuracy would require considering many parameters and source of variation to make a robust conclusion on accuracy[45], but leaves each model on each dataset to have potentially highly different parameters. This obfuscates our true goal[46], which is to understand the impact of the HRR modification in isolation. For this reason we hold as many other variables as constant (depth, layers, neurons per layer, etc) and stick to the defaults found to work well in prior work for non-HRR networks. This allows us to isolate the HRR's impact, and intrinsically puts the HRR network at a disadvantage because it has fewer parameters.

## 5.3  XML HRR Accuracy

We first investigate how the smaller HRR space of $d' \ll L$ dimensions impacts each model's accuracy. Table 1 shows the performance of HRR approach to its respective baselines, evaluated at $k = 1$ for brevity. For all datasets, dimension size $d$ is $400$ except in case of Amazon-13K, Wiki10-31K,

Delicious-200K and Amazon-670K where $d$ is 3000. In many cases our HRR based model approaches the high accuracies of state-of-the-art work, which is informative in that our HRR approach can be implemented with 22 lines of code, compared to hundreds of lines of the more involved methods. We underline the datasets that were used in these prior papers and use $^a$ to denote being within 5% (relative) of scores reported by [35], and $^b$ for 5% of the original AttentionXML [41].

Table 1: Accuracy of our baseline models and their HRR counterparts with the same network architecture otherwise. Cases where the HRR outperforms its baseline counterpart are in **bold**.

| | Bibtex | | Delicious | | Mediamil | | Amazon-12K | |
|---|---|---|---|---|---|---|---|---|
| Model | FC | HRR-FC | FC | HRR-FC | FC | HRR-FC | CNN | HRR-CNN |
| P@1 | 46.4 | **60.3** | 65.0 | **66.5** | 84.8 | 83.9 | 89.1 | 84.5 |
| PSP@1 | 32.5 | **45.6** | 64.2 | **30.0** | 64.2 | 63.7 | 49.2 | 44.2 |
| | | | EURLex-4K | | | | Amazon-13K | |
| Model | FC | HRR-FC | CNN | HRR-CNN | LSTM | HRR-LSTM | FC | HRR-FC |
| P@1 | 73.4 | **77.2**$^a$ | 47.1 | **50.0** | 63.0 | **70.4** | 93.0$^a$ | **93.3**$^{a,b}$ |
| PSP@1 | 32.0 | 30.7 | 18.0 | 17.5 | 26.4 | **26.8** | 52.6 | 49.6 |
| | | | Wiki10-31K | | | | Amazon-13K | |
| Model | FC | HRR-FC | CNN | HRR-CNN | LSTM | HRR-LSTM | LSTM | HRR-LSTM |
| P@1 | 80.4$^a$ | **81.1**$^{a,b}$ | 60.0 | **74.3** | 83.5 | **85.0** | 90.0 | **93.4** |
| PSP@1 | 9.46 | 9.19 | 10.4 | 9.88 | 10.6 | 10.5 | 48.7 | **48.8** |
| | Delicious-200K | | | Amazon-670K | | | | |
| Model | FC | HRR-FC | FC | HRR-FC | CNN | HRR-CNN | | |
| P@1 | 21.8 | **44.9** | 34.6$^a$ | 19.9 | 14.1 | 6.11 | | |
| PSP@1 | 10.5 | 6.84 | 5.22 | **8.45** | 9.39 | 1.51 | | |

Each bold case shows HRR improving over its baseline. HRR-FC Bibtex improved $1.30 - 1.40\times$ and Delicious-200k's P@1 improved $2.06\times$ despite a $1.52\times$ decrease in PSP@1. HRR-CNN improved the Wiki10-31K results by $1.24\times$. In most cases when the HRR results are not better, they are near the performance of the baseline. Most PSP@1 show a $\leq 1$ point decrease, and relative decrease on P@1 results is minimal. The overall results indicate that the HRR approach is moderately worse for retrieval of low-frequency labels, but often significantly better at high-frequency labels (a trade off that is task specific). The primary outlier in behavior is the Amazon-670K dataset, which appears to be pushing the limits of the HRR approach's ability to distinguish between the 670K label vectors. While this is a negative result in terms of XML, our goal was to improve HRRs: the ability to predict & retrieve the a correct label 19.9% of the time out of a space of 670K labels is a near three order of magnitude improvement over the naive HRR without our projection step, which could only perform accurate retrievals of fixed and known vectors in a space of $< 10$ total outputs.

The high relative performance of our HRR-FC compared to XML-CNN and AttentionXML, and our difficulty replicating XML-CNN when fixing the test-set validation bug, does pose interesting questions about the impact of model choice on XML benchmarks. Similar results have been found in systematic review & replication of work in information retrieval and triplet learning [47, 48], but such an investigation is beyond the scope of our work.

## 5.4 HRR Model Compression & Runtime

Because of the size of the output space $L$, the output fully connected layer can represent an enormous amount of the weights within a neural network in this problem space. We can see in Table 2 that the HRR approach can reduce the number of parameters in the output layer by 59-99%, which accounts for 29-42% of the model's total parameters in most cases. The mild exception is the Amazon-13k corpus, which has a large number of input features and comparatively smaller output space. This shows that

Table 2: For each dataset the percentage reduction in parameters of the output layer, and the resulting change for the entire network, by replacing the output layer from a fully-connected softmax with our HRR approach.

| Dataset | Dim $d'$ | % Compression | |
|---|---|---|---|
| | | Output | Network |
| Delicious | 400 | 59.30 | 29.22 |
| EURLex-4K | 400 | 89.98 | 37.80 |
| Wiki10-31K | 3000 | 90.25 | 29.49 |
| Amazon-13K | 3000 | 77.49 | 4.74 |
| Delicious-200K | 3000 | 98.53 | 41.88 |
| Amazon-670K | 3000 | 99.55 | 42.09 |

our HRR approach naturally provides a more compact representation that could be applicable for situations with a high arity of discrete outputs, yet allows learning in a continuous fashion without any sampling tricks.

Fewer parameters leads to a natural speedup in training time too. Table 3 shows the optimization time for a training epoch for a multi-label baseline and HRR-FC where we see an up-to $6.3\times$ reduction in optimization time. The larger the output label space is relative to the rest of the network, the more advantage we obtain. The four cases where HRRs are slower are all the fastest cases, with the smallest dimensions, which prevent the overheads of the HRR implementation from fully realizing their benefit. This is because the cross-entropy calculations are highly optimized, and faster than the cosine similarity and for loops needed by our HRR implementation.

Table 3: Speedup in training time for HRR-FC over the FC model. **Bold** indicates improved runtime.

| Dataset | Speedup |
|---|---|
| Bibtex | 0.35 |
| Delicious | 0.38 |
| Mediamill | 0.39 |
| EURLex-4K | 0.35 |
| Wiki10-31K | **1.48** |
| Amazon-13K | **1.33** |
| Delicious-200K | **4.47** |
| Amazon-670K | **6.28** |

We note that separate from the training time, at inference time on the test set the HRR approach currently takes the same time as the fully-connected output. This is because each positive label $c_p$ is queried against the network's output $\hat{s}$, but this need not be the case. There is further potential for improvement by developing specialized Maximum Inner Product Search (MIPS) kernels that interface nicely with PyTorch, don't require memory copies, and can re-constitute the output labels as needed. These are all engineering challenges we consider beyond scope of the current work, as we are concerned with how to learn through HRRs.

## 5.5 Assessing Impact of Hyper-parameters

By demonstrating non-trivial learning that can match or even improve upon the accuracy of a standard fully-connected output layer that naturally leads to reductions in model size/memory use and improved training time as the models get larger, we have shown that our projection step makes HRRs viable for future learning tasks. This is because without our modification we get consistent random-guessing performance across all datasets.

However, our larger goal is to gain new information about how to learn with HRRs. As such we now perform a number of ablation experiments to elucidate potential factors relevant to future work outside of XML. This includes the impact of the output dimension $d'$, the effect of increasing model size, and if the HRR vectors should be altered via gradient descent or left static as we have done. Due to space limitations, we will show many results on just the Wiki10-31K dataset in this section as we found it tracks well with overall behavior on other corpora.

**Label Dimension Size:** While training a model with a dense label representation, the vector size $d'$ of the label is fixed. Figure 2 shows the impact of vector dimension size on the model's precision. As observed, Precision@5 increases when the dimension size is increased (and is representative of other metric's behavior). For datasets *Wiki10-31K* and *AmazonCat-13K*, the precision begins to plateau when the dimension size is approximately 10% of the number of labels in that dataset. The trend is consistent across all Precision@k measurements. For a larger dataset like *Delicious-200K*, the precision levels off at

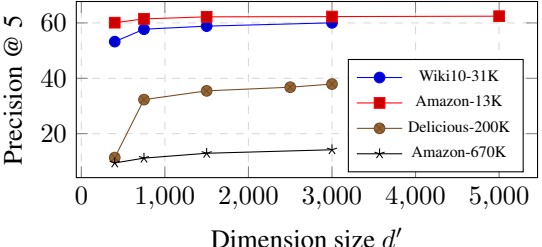

Figure 2: P@5 (y-axis) performance quickly plateaus as the output network size $d'$ (x-axis) increases. The plateau occurs at $d' \leq 10\% \cdot L$, requiring a smaller percentage of all labels $L$ as $L$ increases.

a dimension size of 2500, which is 1.25% of the number of labels. Thus, Figure 2 suggests that HRR can substantially compress the size of the label vector as the number of labels increases (and with it the label distribution itself is increasingly skewed).

**Network Layer Size:** Our initial results in §5.3 showed that our HRR-FC & HRR-CNN have comparable or better accuracy, while also having less parameters. We also looked at how these results would change as both models are made larger. Results from Wiki10-31K are

shown in table 4. We can see that even moderate changes to the baseline network actually produce a drop in accuracy, which is expected since we tuned the initial architecture to baseline performance. For our HRR-FC more layers or hidden neurons have minor impact, but combined with larger output size $d'$ can increase performance from the initial 53.25% up to 58.51%. All of these HRR-FCs still have fewer parameters than the initial baseline network.

**Updating $p$ & $m$ Vectors:** The $p$ and $m$ vectors are the key to the cumulative positive and negative class vectors. It is possible to backpropagate to these vectors and alter them during training, rather than hold them fixed as we have done. We measure the impact of updating $p$ and $m$ vectors in comparison to keeping them static while the model is being trained. We found that maintaining $p$ and $m$ vectors fixed is beneficial when the dimension size is small ($d' \leq 400$). As $d'$ increases we observe no impact from learning the values of $p$ and $m$. This was consistent across all datasets and architectures.

Table 4: Increasing the number of layers or hidden neurons has less benefit to the FC baseline compared to the HRR approach.

| Net. | Layers | Hidden $h$ | Out $d'$ | P@5 |
|------|--------|-----------|----------|-----|
| FC | 2 | 512 | 30938 | 46.64 |
| | 2 | 2048 | 30938 | 47.91 |
| | 3 | 512 | 30938 | 41.92 |
| | 3 | 2048 | 30938 | 45.25 |
| HRR | 3 | 512 | 400 | 55.06 |
| | 3 | 2048 | 400 | 55.36 |
| | 3 | 2048 | 750 | 56.84 |
| | 3 | 2048 | 1500 | 57.80 |
| | **3** | **2048** | **3000** | **58.51** |

**Updating Label Vectors $c_p$:** Another question is if we should have allowed the label vectors $c_p$ to be altered during training, rather than holding them fixed at initialized values. We test this in Table 5 using a network with multiple different numbers of hidden neurons $h$ and output dimensions $d'$ for a 2-layer network, where the $\nabla_{c_p}$ is the performance when we allow the concept vectors to be learned. We note that this was implemented by always back-propagating through our complex unit magnitude projection $\pi$, and resulted in no significant performance difference. Experiments without the projection $\pi$ had significantly degraded results that gave random-guessing performance.

Table 5: Ablation test of whether label vectors $c_p$ should be altered during training (right most column) or remain fixed (2nd from right). Results are from Wiki10-31K as a representative dataset.

| | | P@5 | |
|-----------|----------|-------|-----------------|
| Hidden $h$ | Out $d'$ | Fixed | $\nabla_{c_p}$ |
| 512 | 400 | 55.06 | 53.50 |
| 2048 | 400 | 55.36 | 55.45 |
| 2048 | 750 | 56.84 | 56.75 |
| 2048 | 1500 | 57.80 | 57.69 |
| 2048 | 3000 | 58.51 | 58.18 |

These experiments show that altering the HRR vectors, at least in the XML application, have no benefit. We had expected this to perform better by giving the network a chance to adjust for any unfavorable initialization. Our current hypothesis is that the network's own parameters are sufficient to learn how to leverage the dynamics of HRR operations, which should still encourage the same properties while training, making the weights of the HRR concept vectors redundant. Further exploration of this is needed in future studies.

# 6 Conclusion

We have improved the initialization of HRRs to increase binding and retrieval accuracy, allowing for a convenient & differentiable neuro-symbolic approach. To demonstrate potential utility we have applied HRRs to extreme multi-label classification. We observe reduced model size, faster training, and stable-to-improved accuracy. This provides evidence for the general utility and further study of HRRs. The learning dynamics of HRRs are not yet fully understood by our work and have the counter-intuitive behavior that learning with them, but not altering them, tends to have the best results. Compared to prior works looking at HRRs we have significantly improved the viability of building more complex networks that leverage the symbolic manipulation properties of HRRs, which opens the door to a new kind of approach to modeling problems.

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
