## A  A Note on Applications and Future Work

The applications of HRRs may not be immediate, given the approach has been out-of-vogue amongst most machine learning practitioners for many years. Long term we believe improvements in neuro-symbolic learning are important for better generalization of ML methods to novel inputs and situations, as argued by [3]. In the short term future, we do believe HRRs may have considerable opportunity to provide enhancements. Transformers via their "query, key, value" Multi-Headed Attention (MHA) are a natural place to explore HRRs due to the match of logical design, while potentially avoiding MHA's high costs and are supported by similarly motivated analysis by Schlag et al. [4] through the lens of an associative memory. Similar recent works on TPR augmented RNNs for natural language processing (NLP) [5, 6] show value to endowing modern designs with classic symbolic-connectionist ideas. The same inspiration and other neuro-symbolic work on question-answering with TPRs [6] leads us to believe HRRs maybe have similar potential for such systems, and in particular as a way to extract, or augment the knowledge base of an a queryable system in a way that current methods do not yet allow. Broadly we believe many NLP applications of HRRs may exist given the common need to perform binding of subjects to their associated nouns, entity resolution, and the large variety of binding like problems that occur across NLP tasks. Ultimately, we hope that the most interesting work will come from taking new perspectives on how loss functions and problems may be modeled, as we have done in § 4, to enable new kinds of tasks and applications.

## B  Understanding Compositional Representations with HRR

In this section, we provide an illustrative example of how a compositional representation can be constructed with holographic reduced representations. As shown in Fig. 3, a dog is represented a combination of the different parts of its body. The representation is in the form of a tree and consists of a two-level hierarchy where the *head* part is further represented as a combination of *eyes*, *nose* and *mouth*. Our objective is to create design a dense vector representation that can represent this hierarchy. There are multiple ways in which a representation can be constructed, such as a binary format, or concatenating individual attribute vectors representations. HRRs allow us construct a dense representation that can be decomposed while maintaining the vector dimension size $d$ constant.

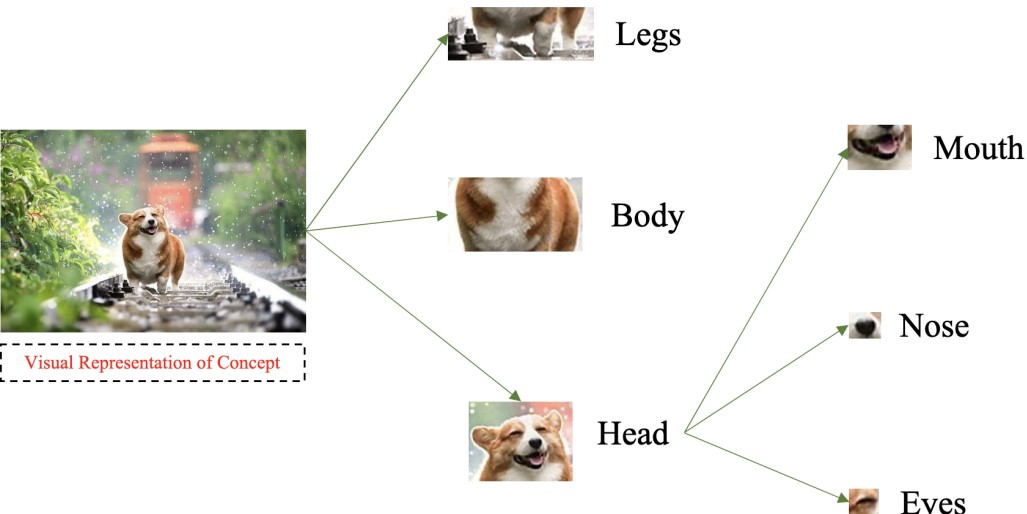

Figure 3: Example representation of *dog* as a combination of different parts. The representation is a two level hierarchy where its *head* can be subdivided into components.

Fig. 4 shows how HRR can be utilized. As described in §1, each attribute is represented as combination of two vectors: a key (k) and attribute vector. The $k^{\dagger}$ is used to retrieve the original attribute vector.

In the given example, the trace for *dog* (final dense representation) is computed by adding all $key \otimes attribute$ pairs. We ask the query: **Do dogs have legs?** and retrieve the attribute for *legs* by

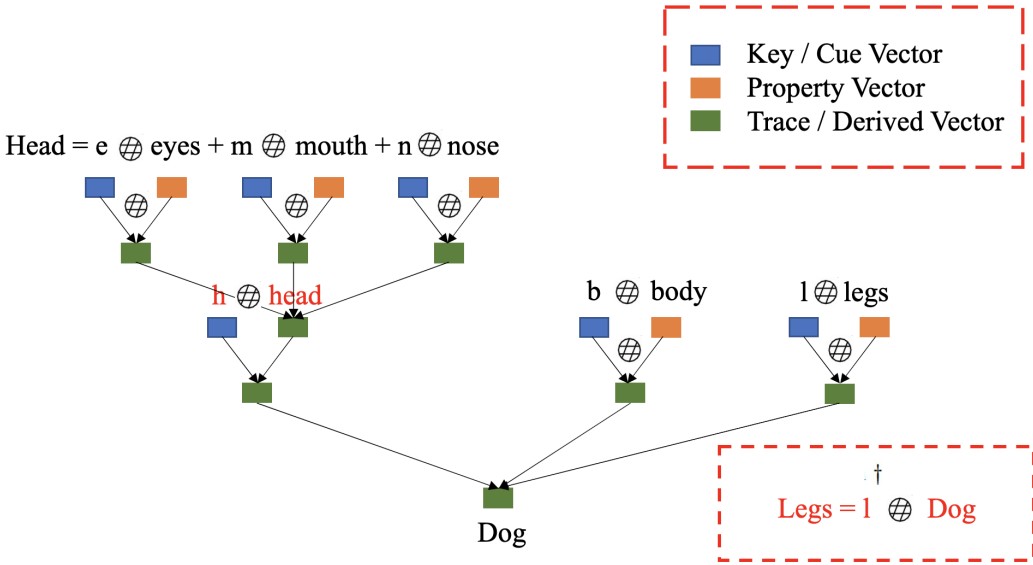

Figure 4: Following is the vector representation of a *dog* using HRR. There are three types of vector representations, namely, the **Key / Cue** vector, the **attribute** or **property** vector and **trace / derived** vector. Trace vectors can be added to form a combined vector representation. The diagram shows how a two-level hierarchy is represented with HRR. **It is important to note that the dimension size of the representation remains constant**. To query the vector, the inverse of the key for a given attribute is utilized (with the unbind operator). In this example, we ask the question: **Does a dog have legs?**

computing $l^\dagger \otimes Dog$ where $l$ is the key for the attribute *legs* and $Dog$ is the trace vector representing the concept. A simple yes/no response can then be obtained by comparing to the property, via $legs^\top l^\dagger \otimes Dog$ Because the $\otimes$ operation is associative and communicative, we can also ask if dogs have eyes by checking $eyes^\top e^\dagger \otimes Dog$, though a stronger response will be obtained by using the full structure of the encoding and checking $eyes^\top h^\dagger \otimes e^\dagger \otimes Dog$.

The reader may then ask, why should the HRR operation allow us to answer queries like $legs^\top l^\dagger \otimes Dog$ in a fixed dimensional space? As an example we will reproduce the excellent illustrative worked example by Plate [1], followed by a new derivation showing the same nature when distractor terms are included in the statement.

Consider a $d = 3$ dimensional space, where we wish to compute $c^\dagger \otimes (c \otimes x)$, we will get the result that:

$$c^\dagger \otimes (c \otimes x) = \begin{bmatrix} x_0 \left(c_0^2 + c_1^2 + c_2^2\right) + x_1 c_0 c_2 + x_2 c_0 c_1 + x_1 c_0 c_1 \\ \quad + x_2 c_1 c_2 + x_1 c_1 c_2 + x_2 c_0 c_2 \\ x_1 \left(c_0^2 + c_1^2 + c_2^2\right) + x_0 c_0 c_1 + x_2 c_0 c_2 + x_0 c_0 c_2 \\ \quad + x_2 c_1 c_2 + x_0 c_1 c_2 + x_2 c_0 c_1 \\ x_2 \left(c_0^2 + c_1^2 + c_2^2\right) + x_0 c_0 c_1 + x_1 c_1 c_2 + x_0 c_1 c_2 \\ \quad + x_1 c_0 c_2 + x_0 c_0 c_2 + x_1 c_0 c_1 \end{bmatrix} \tag{10}$$

There are two simplifications that can be done to this resulting matrix by exploiting the fact that all elements in the matrix are sampled according to the distribution $x \sim \mathcal{N}(0, \frac{1}{d})$. First there is the pattern $x_i \cdot \prod_{j=0}^{d-1} c_j^2$ on the left hand side. The sum of squared normals will in expectation be equal to 1, but we will subtract that value in our change of variable to create $\xi = \left(c_0^2 + c_1^2 + \cdots + c_d^2\right) - 1$, which will then have the distribution $\xi \sim N\left(0, \frac{2}{n}\right)$. Second, the right-hand side will have $d(d-1)$ products

of the form $x_i c_j c_k, \forall j \neq k$. Summing all of these into a new variable $\eta_i$ products $\eta_i \sim \mathcal{N}\left(0, \frac{d-1}{d^2}\right)$. Inserting $\xi_i$ and $\eta_i$ we get:

$$\boldsymbol{c}^\dagger \otimes (\boldsymbol{c} \otimes \boldsymbol{x}) = \begin{bmatrix} x_0(1+\xi) + \eta_0 \\ x_1(1+\xi) + \eta_1 \\ x_2(1+\xi) + \eta_2 \end{bmatrix} = (1+\boldsymbol{\xi})\tilde{\mathbf{x}} + \tilde{\boldsymbol{\eta}} \tag{11}$$

Since both $\xi_i$ and $\eta_i$ have a mean of zero, we get the final result that $\mathbb{E}[\boldsymbol{c}^\dagger \otimes (\boldsymbol{c} \otimes \boldsymbol{x})] = \boldsymbol{x}$, allowing us to recover a noisy approximation of the original bound value. The communicative and associative properties of the HRR's construction then extend this to the more complex statements that are possible, and accumulate the noise of the resulting variables.

To demonstrate this, we will perform another example with $\boldsymbol{c}^\dagger \otimes (\boldsymbol{c} \otimes \boldsymbol{x} + \boldsymbol{a} \otimes \boldsymbol{b})$. This will be performed with $d = 2$ in order to avoid visual clutter, and results in the equation:

$$
\begin{aligned}
\boldsymbol{c}^\dagger \otimes (\boldsymbol{c} \otimes \boldsymbol{x} + \boldsymbol{a} \otimes \boldsymbol{b}) &= \begin{bmatrix} \frac{c_0(a_0b_0+a_1b_1+c_0x_0+c_1x_1)-c_1(a_0b_1+a_1b_0+c_0x_1+c_1x_0)}{(c_0-c_1)(c_0+c_1)} \\ \frac{c_0(a_0b_1+a_1b_0+c_0x_1+c_1x_0)-c_1(a_0b_0+a_1b_1+c_0x_0+c_1x_1)}{(c_0-c_1)(c_0+c_1)} \end{bmatrix} \\
&= \begin{bmatrix} \frac{a_0b_0c_0-a_0b_1c_1-a_1b_0c_1+a_1b_1c_0+c_0^2x_0-c_1^2x_0}{c_0^2-c_1^2} \\ \frac{-a_0b_0c_1+a_0b_1c_0+a_1b_0c_0-a_1b_1c_1+c_0^2x_1-c_1^2x_1}{c_0^2-c_1^2} \end{bmatrix}
\end{aligned} \tag{12}
$$

Notice that the red highlighted portion of the equation is the product of independent random variables, meaning two important properties will apply: $\mathbb{E}[XY] = \mathbb{E}[X] \cdot \mathbb{E}[Y]$ and $\mathrm{Var}(XY) = \left(\sigma_X^2 + \mu_X^2\right)\left(\sigma_Y^2 + \mu_Y^2\right) - \mu_X^2 \mu_Y^2$. Because these random variables have a mean $\mu = 0$, the products result in a new random variable with the same mean and reduced variance as the original independent components. The first property gives

$$\mathbb{E}[XY] = \mathbb{E}[X] \cdot \mathbb{E}[Y] = 0 \cdot 0 = 0$$

and the second property gives:

$$\mathrm{Var}(XY) = \left(\sigma_X^2 + \mu_X^2\right)\left(\sigma_Y^2 + \mu_Y^2\right) - \mu_X^2 \mu_Y^2 = \left(\left(\frac{1}{d}\right)^2 + 0\right)\left(\left(\frac{1}{d}\right)^2 + 0\right) - 0 = \frac{1}{d^4}$$

That reduces each product of $a_i b_j c_k$ into a new random variable with a mean of zero, and then the sum of these random variables, due the the linearity of expectation, will be a new random variable with an expected value of zero. So in expectation, the highlighted red terms will not be present (but their variance due to noise will cause errors, though the variance is harder to quantify due to reuse of random variate across the products). Thus we get the expected result of:

$$\begin{bmatrix} \frac{c_0^2 x_0 - c_1^2 x_0}{c_0^2 - c_1^2} \\ \frac{c_0^2 x_1 - c_1^2 x_1}{c_0^2 - c_1^2} \end{bmatrix} = \begin{bmatrix} x_0 \\ x_1 \end{bmatrix} \tag{13}$$

Which recovers the original $\boldsymbol{x}$ value that was bound with $\boldsymbol{c}$, even though additional terms (e.g., $\boldsymbol{a} \otimes \boldsymbol{b}$ are present in the summation.

## C    Implementation

Our implementation for all experiments is included in the appendix, and is based off original XML projects from the authors of AttentionXML and XML-CNN, and as such contain significant code that is specific to the data loaders, their original training pipelines, and other features extraneous to the task of understanding just the code for an HRR. As such we take a moment to demonstrate the PyTorch code one could write (as of 1.8.1 which added revamped support for complex numbers and ffts) to implement our HRR approach.

First are the operations for binding, the inverse and approximate inverse functions, and our projection step. This can be accomplished in just 10 lines of Python code, as the below block shows. The use of the *real* and *nan_to_num* functions are defensive guards against numerical errors accumulating in the fft functions that could cause small complex values to occur in the results of computations.

```python
def binding(x, y):
    return torch.real(ifft(torch.multiply(fft(x), fft(y))))
def approx_transpose(x):
    x = torch.flip(x, dims=[-1])
    return torch.roll(x, 1, dims=-1)
unbind = lambda x, y: binding(s, approx_transpose(y))
def projection(x):
    f = torch.abs(fft(x)) + 1e-5
    p = torch.real(ifft(fft(x) / f))
    return torch.nan_to_num(p) #defensive call
```

The loss is also easy to implement, and below we show a slice of how most of our models implemented the loss approach. The `inference` function takes in a `p_or_m` variable that is either the present vector $p$ or the missing vector $m$, extracts the target vector from the prediction (i.e., $\mathbf{p}^* \otimes \hat{\mathbf{s}}$ or $\mathbf{m}^* \otimes \hat{\mathbf{s}}$), and then L2 normalizes the result so that the down-stream dot product becomes equivalent to the cosine distance. The `inference` function is then used for computing J_p and J_n, but using the abs function instead of the true angular distance as a micro optimization. We obtain the same results regardless of that implementation choice, but the abs calls are just a bit faster to run and avoid add inverse cosine calls.

```python
def inference(s, batch_size, p_or_m):
    vec = p_or_m.unsqueeze(0).expand(batch_size, self.label_size) #make
    ↪   shapes work
    y = unbind(s, vec) #(batch, dims), extracting the target values from
    ↪   prediction
    y = y / (torch.norm(y, dim=-1, keepdim=True) + 1e-8) #normalize so that
    ↪   results will be cosine scores
    return y

convolve = inference(s, target.size(0), p)
cosine = torch.matmul(pos_classes, convolve.unsqueeze(1).transpose(-1,
↪   -2)).squeeze(-1) #compute dot products
J_p = torch.mean(torch.sum(1 - torch.abs(cosine), dim=-1))

convolve = inference(s, target.size(0), m)
cosine = torch.matmul(pos_classes, convolve.unsqueeze(1).transpose(-1,
↪   -2)).squeeze(-1)#compute dot products
J_n = torch.mean(torch.sum(torch.abs(cosine), dim=-1))

loss = J_n + J_p # Total Loss.
```

As seen in the implementation above, J_p and J_n are the positive and negative losses. The `cosine` value can be positive or negative value ranging from 1 to $-1$. While inferring if an `unbind` vector is related to a label vector, we compute the `cosine` distance. Hence, while computing the loss, we take the absolute value of the cosine in order to maintain the positive loss minimizing towards 0.

## D   Binding Capacity and VSA Selection

HRRs are but one of many possible vector symbolic architectures (VSAs) that one could select. For the purposes of our work, we had four desiderata.

1. The VSA vectors should naturally exist in the reals, since most deep learning applications are using real-valued vectors.

2. The VSA should be composed entirely of differentiable operations, so that learning may be possible.

3. The VSA should be of minimal additional overhead.

4. The VSA should be as effective as possible at the binding operation.

The first two of these items are binary requirements that a VSA either has or not. This excludes many VSAs that operate in the complex domain or discrete spaces, leaving us with three potential candidates: HRRs, continuous Multiply-Add-Permute (MAP-C, distinguishing from its binary alternative)[49][50], and the most recently developed Vector-derived Transformation Binding (VTB). Of these three the MAP-C option is least desirable because it requires a clipping operation to project vectors values into the range of $[-1, 1]$, which results in sub-gradients and zero-gradient values that will make optimization more challenging.

In evaluating the overhead of each method, HRRs and MAP-C are satisficing, they are both composed of operations well defined and optimized by existing deep learning systems. The VTB approach requires a sparse block-diagonal Kronecker product that we found is not well optimized in current tools, often requiring $10\times$ the memory to back-propagate through compared to the HRRs and MAP-C, making it less desirable. We stress we do not believe this to be a fundamental limitation of VTB, but a limitation of current tooling. We are confident a custom implementation will work without memory overheads, but wish to constrain ourselves to already existing functions of PyTorch due to simplicity and expediency.

### D.1 Capacity For Error Free Retrieval

The last question, VSA effectiveness, then becomes part of the decision process in selecting a final VSA to use. To help elucidate how we came to chose the HRR, we will discuss experimental results on the capacity of the VSAs with respect to problems of the form:

$$S = \sum_{i=1}^{n} \text{bind}(\boldsymbol{x}_i, \boldsymbol{y}_i)$$

This form of $S$ is the same that we rely on to develop our loss framework in § 4, and does not capture all the ways that a VSA may be used. This analysis is thus not conclusive to holistic VSA effectiveness, but it does capture the common form of capacity that we will discuss that influenced our selection.

To estimate the capacity, after $n$ pairs of items are bound together we attempt to unbind $y_i$ which should return $\text{unbind}(S, \boldsymbol{y}_i) = \hat{\boldsymbol{x}}_i \approx \boldsymbol{x}_i$. There will then be a pool of $n$ random distractor vectors $\boldsymbol{z}_1, \ldots, \boldsymbol{z}_n$, sampled in the same manner used to construct the $\boldsymbol{x}_i$ and $\boldsymbol{y}_i$ values of the VSA being tested. If there exists any $\boldsymbol{z}_j$ such that $\text{cos-sim}(\hat{\boldsymbol{x}}_i, \boldsymbol{x}_i) < \text{cos-sim}(\hat{\boldsymbol{x}}_i, \boldsymbol{z}_j)$, then that $j$'th item is considered to be incorrectly retrieved. So our capacity will be the value of $n$ such that no more than $t$ retrieval errors occur.

Figure 5 shows the capacity of each method given a threshold of no more than 3% error, as estimated by 10 trials of randomly selecting all $n$ pairs and distractor items, with $n$ tested at values of $\sqrt{2}^j$. We can see that the naive HRR actually has the worst performance, in part due to its numerical instability/approximation error. It is also important to note that the HRR's original theory developed by Plate [1] states that the capacity should grow linearly with the dimension size. We find for naive HRRs this is not the case.

Because HRRs did best satisfying all requirements but the capacity issue, we chose to attempt to improve the HRRs so that they would be more effective[4]. As we discussed in § 3.1 this can be done with our projection operation, which restores the theoretically expected behavior of linear capacity improvement with dimension size $d$, and brings HRRs to parity with the best performing (in terms of capacity) VSA the VTB. Since the HRR required significantly less memory than VTB, and was slightly faster in our testing, our improved HRR became the most logical choice to move forward with.

---

[4]This work in fact started before the VTB method was published, but was reconsidered when we learned of it.

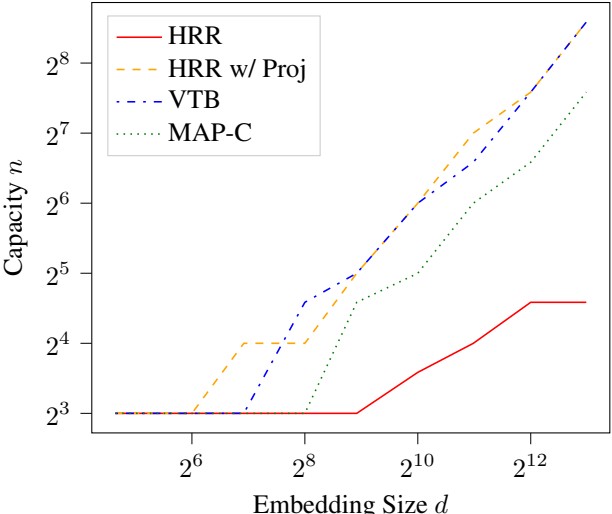

Figure 5: Capacity based on the ability to represent $n$ items bound together and, compared to $n$ false distractors, correctly identify the true item as the most-similar. As the dimension of the embedding space $d$ increases, most VSAs capacity increases linearly.

For further edification about the capacity of each evaluated approach, we show in Figure 6 the probability of a retrieval error occurring as the number of $n$ items increases, with the standard deviation over 10 trials shown in the highlighted region. As can be seen, our improved HRRs and VTB have statistically indistinguishable performance, which was quite surprising, and may lead to further theory work around the limits of binding capacity in a fixed-length representation.

In all cases we can see that while capacity at a threshold $t$ does increase linearly with dimension $d$ for the non-HRR approaches[5]. It is also worth noting that capacity is a fairly hard limit, with error increasing slowly until the capacity is reached, at which point the probability of error begins to increase rapidly with expanded set size. There are also other forms of VSA capacity that are beyond our current scope, especially when discussing mechanisms like RNNs built from VSA [51]. Our results should not be taken as conclusive holistic descriptions of HRRs vs other VSAs, but are limited to the form of capacity we have discussed in this section and is most relevant to our application.

In relation to our results in storing tens to hundreds of thousands of vectors, we note that our results in § 5 are based on learning to extract the correct objects, and the penalty term is based on a single averaged representation of all other concepts, which thus down weights any false-positive response due to noise of a single item. The capacity results we discuss in this section are with respect to *any* of the original $z_i$ distractors having a higher response, which requires $n$ brute force evaluations and is a harder scenario than what we required.

### D.2 Capacity For Average Response Range

The capacity question we have just walked through is for error free recognition of the true item as more similar than a set of $n$ distractors. However, our use of the HRR operations poses a mixed representation. In $J_p$ we perform extraction of the classes present, but $J_n$ relies on the average response value being accurate. $J_p$ requires on average less than 76 explicit items to be retrieved in all our datasets, but $J_n$ is representing the average response over tens to hundreds of thousands of items. So while $J_n$ requires a "larger" capacity in some sense, it only requires the average response to be stable.

We can explore this in our data by looking at Figure 7, where we plot the mean and standard deviation of *individual* responses. The solid lines correspond to the same results as presented in Figure 1, but

---

[5]It is possible naive HRRs would increase linearly given even larger values of $d$, but experimentation past that point is unreasonable.

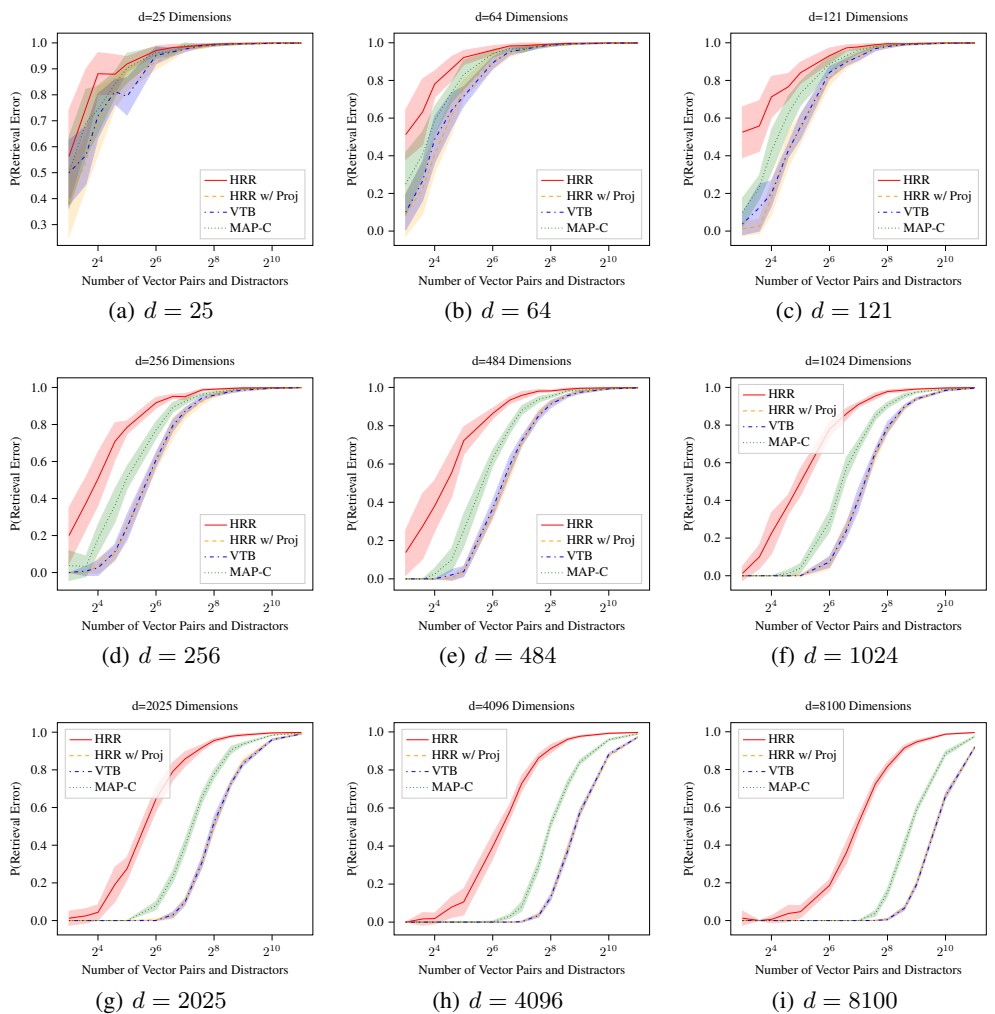

Figure 6: Probability of a retrieval error (y-axis) when, given $n$ (x-axis) pairs of objects bound together and $n$ distractor items, the unbound concept vector is more similar to a distractor than the true original object. The values of $d$ are prefect squares due to a technical requirement of the VTB approach.

we are looking at only the improved HRR, and showing the standard deviation of the individual responses that form the average.

Given this results there are multiple ways we could look at when the HRR response begins to "fail". If we look at when do the mean and standard deviation of the responses start to overlap for the present/absent cases, that starts around $n = 512$ items. If we look at when the standard deviation starts to approach the other item's mean, that occurs around $n = 1,536$. If we look at when the mean response begins to deviate away from the target value of 1/0, that does not start to occur until around $n = 2^{16} = 65,536$ (and is still very close, but larger values of $n$ are computationally expensive)! This stability of the average for large $n$ is an important component of our loss component $J_n = \cos\left(\mathbf{m}^\dagger \otimes \hat{\mathbf{s}}, \sum_{\mathbf{c}_p \in \mathcal{Y}^p} \mathbf{c}_p\right)$ is implicitly working over an average response of all the negative labels.

This shows that the distributional average around the desired response value for present/absent items is very stable, but the tails of the distribution do begin to grow as you try to pack more and more into the single representation. This validates further why we need to use a normalized response via

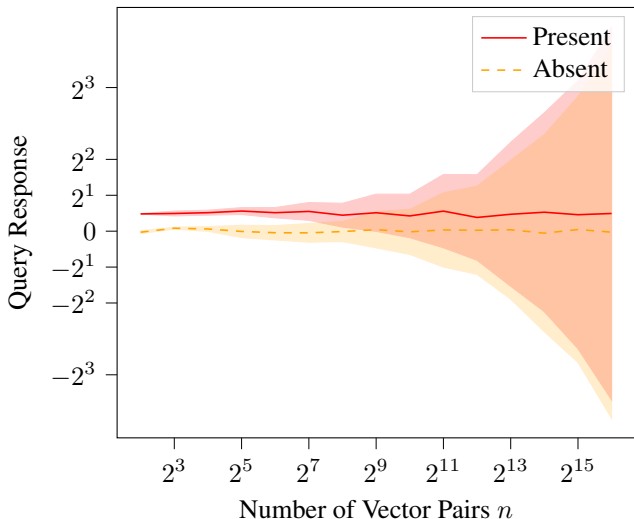

Figure 7: Distribution of the response to queries of the form $q^\dagger \sum_{i=1}^n a_i \otimes b_i$ for cases where $q$ is present or absent from the summation. The standard deviation of responses is shown in the shaded region, y-axis is symlog scale and x-axis is log scale.

the cosine similarity when extracting the present terms, but also how the $J_n$ term can function well despite the large symbolic query space.

## E   Datasets

All datasets and their source are given in Table 6.

Table 6: **Dataset Statistics** from Bhatia et al. [44]. The table describes the statistics of each dataset utilized for experiments and includes the number of features per sample, number of labels in each input sample, the diversity of the dataset represented through the average number of points per label and average number of labels in each sample.

| Dataset | Features | Labels | Avg. Samples per Label | Avg. Labels per Point |
|---|---|---|---|---|
| Mediamill [52] | 120 | 101 | 1902.15 | 4.38 |
| Bibtex [53] | 1836 | 159 | 111.71 | 2.40 |
| Delicious [54] | 500 | 983 | 311.61 | 19.03 |
| EURLex-4K [55] | 5000 | 3993 | 25.73 | 5.31 |
| Wiki10-31K [56] | 101938 | 30938 | 8.52 | 18.64 |
| Ama13K [57] | 203882 | 13330 | 448.57 | 5.04 |
| Delicious-200K [58] | 782585 | 205443 | 2.29 | 75.54 |
| Amazon-670K [55] | 135909 | 670091 | 3.99 | 5.45 |

## F   Additional Metrics

Next there is the DCG@k and PSDCG@k scores, which differ only by the inclusion of the $p_l$ term being absent / present respectively. PSDCG is shown below.

$$\text{PSDCG@}k := \sum_{l \in \text{rank}_k(\hat{\mathbf{y}})} \frac{\mathbf{y}_l}{p_l \log(l+1)}$$

As recommended we use the normalized versions of each giving us nDCG@k and PSnDCG@k, resulting in eq. (14) and eq. (15).

$$\text{nDCG@}k := \frac{\text{DCG@}k}{\sum_{l=1}^{\min(k,\|\mathbf{y}\|_0)} \frac{1}{\log(l+1)}} \tag{14}$$

$$\text{PSnDCG@}k := \frac{\text{PSDCG@}k}{\sum_{l=1}^{k} \frac{1}{\log(l+1)}} \tag{15}$$

Across all experiments we see that results across different values of $k$ tend to be consistent. The pairings of Precision@k and nDCG@k and PSprec@k and PSnDCG@k are highly correlated in all our results, and equivalent for $k = 1$. For this reason we will show most results at $k = 1$ for brevity, with larger tables of results in the appendix.

## G  Computer Resources

Training was done primarily on a shared compute environment, but in general we had access to only one or two compute nodes at any given time. The main compute node used had a Tesla V100 with 32 GB of RAM, which could barely fit the Amazon-670K experiments during training. Going through all datasets to obtain results took approximately 2-weeks of compute time per model tested, and we have three models under evaluation. Combined with other experiments that did not pan out, we did not have the capacity to perform the 25+ runs that we would prefer to provide robust measures of variance in our results. We do report that spot checking smaller datasets like Bibtex that had large effect sizes consistently returned those large effect sizes.

## H  Inference for XML with HRR

We take a moment to be more explicit about how inference is done with HRRs to perform XML prediction, and also discuss further potential advantages that could be achieved given more software engineering effort.

Given a network's prediction $\hat{s} = f(x)$, inference can be done by simply iterating though all class HRR vectors $c_1, c_2, \ldots, c_L$, and selecting $\arg\max_i c_i^\top p^* \otimes \hat{s}$ to determine that class $i$ is the top prediction of the network. To select the top-$k$ predictions, as is common in XML scenarios, one simply selects the top-$k$ largest dot products to be the predicted set. Or one can use a threshold of $c_i^\top p^* \otimes \hat{s} > 0.5$ to select the set of likely present classes. While this is *not* a calibrated probability, this works out by the math of HRRs that a value being present should produce a dot product of $\approx 1$ and non-present values should produce a dot product of $\approx 0$.

The above describes how inference is currently done in our code. We note that it could be further accelerated. This is because the inference formulation $\arg\max_i c_i^\top (p^* \otimes \hat{s})$ is now a *Maximum Inner-Product Search* (MIPS) problem, for which many algorithms have been designed to accelerate such queries [32, 59, 60]. We have not incorporated these tools due to current freely available software not being well designed for our use case. This appears to be a purely software engineering problem, and beyond our current capacity to implement. For example, the MLPACK3 library[6] has MIPS algorithms that can perform the exact search for the top-$k$ items in expected $\mathcal{O}(\log n)$ time after building the index at cost $\mathcal{O}(n \log n)$. Our setup would allow such a construction, but the library is based on CPU only calculations. For the scale of datasets that are publicly available that we tested, the constant-factor speedup of a GPU is still faster than the $\mathcal{O}(\log n)$ search. If we had access to private XML corpora with 100 million classes[33, 39], we would expect this result to change.

The only software we are aware of with GPU support for *approximate* MIPS search is the FAISS library[7]. While broadly useful, the library does not support the functionality we need to avoid significant overheads that make it slower than a brute force search in this case. First, the FAISS library requires keeping its own copy of all vectors $c_1, \ldots c_L$ in GPU memory. This is a non-trivial

---

[6] https://www.mlpack.org/
[7] https://github.com/facebookresearch/faiss/wiki

cost that can make it difficult for us to fit the model in memory at the same time, which is the case where such MIPS searches would prove advantageous. Our implementation does not require storing the symbols $c_i$ in memory, because they can be re-constructed as needed based on a random seed. This makes the brute force search faster because it requires no additional memory accesses once $p^* \otimes \hat{s}$ has been computed and stored in GPU memory. This makes our brute force considerably faster, and causes the FAISS implementation to have significant overhead for unneeded memory use in its normal index structure combined with explicitly storing all $c_i$.

# I HRR Model Runtime with XML-CNN

In §5.4, we measured the performance of the baseline FFN (FC) and HRR-FFN (HRR-FC) and showed how its execution time decreases as the number of labels increase. The cost of a single forward pass through the network is lower than baseline because the size of the output layer is smaller. Similarly, we analyze the impact of output layer size reduction on the XML-CNN architecture [61]. We observe in table 7 that execution time reduces across larger datasets, but initially the optimization time is higher (amazoncat-12k). The optimization time accounts for both: (a) the time taken to compute the loss and (b) the time taken to calculate the gradients and update the network.

Table 7: **Model Execution & Optimization Time**. Compare execution and optimization time for XML-CNN [61] and HRR-CNN. Execution time is the average time (seconds) to perform a forward pass and inference through the model for 1 epoch of training. It corresponds to the throughput of the model. Similarly, optimization time includes the time to compute the loss and optimize the model. As observed, overall *Train* time reduces as the number of labels in the dataset increases.

| Dataset | Model | Execution Time | Optimization Time |
|---|---|---|---|
| EURLex-4K | CNN | **0.466** | **2.306** |
| | HRR-CNN | 0.467 | 3.657 |
| Wiki10-31K | CNN | **0.630** | **2.665** |
| | HRR-CNN | 0.712 | 3.286 |
| AmazonCat-12K | CNN | 16.722 | **83.305** |
| | HRR-CNN | **16.178** | 117.098 |
| Amazon-670K | CNN | 239.48 | 734.694 |
| | HRR-CNN | **122.665** | **301.376** |

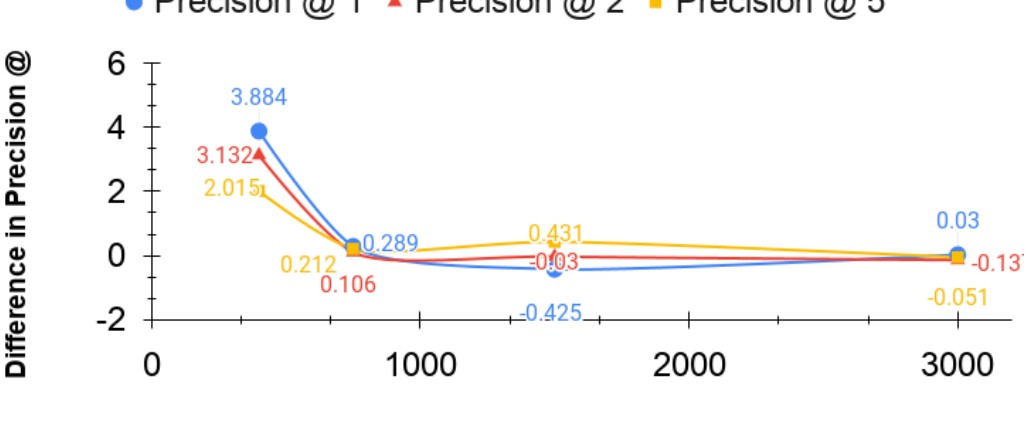

Figure 8: Difference between the precision @ k for HRR-FC trained when $p$ & $m$ vectors are updated. Positive values indicate a preference for fixed values, negative a preference for learned values.