# OpenReview forum: "Learning with Holographic Reduced Representations"
_NeurIPS.cc/2021/Conference — NeurIPS 2021 Spotlight_

### Official Review · Reviewer_sKB5 · 2021-07-07

**Rating:** 8
**Confidence:** 4

**Summary:**

The authors improve Plate's HRRs, which opens up new possibilities for differentiable neuro-symbolic components in neural networks. The authors demonstrate the effectiveness of the introduced projection step in one setting: extreme multi-label problems. The paper is very well written, and the results are convincing with an exemplary analysis.

**Limitations And Societal Impact:**

I have no suggestions for improvement.

**Main Review:**

Neural Networks struggle to generalise in systematic ways similar to humans. For this reason, there has been in recent years a push towards more systematic / neuro-symbolic representations in connectionist models. The binding problem is central in this research direction [1]. However, it is already known since the '90s how one may represent compositional/symbolic structures in a continuous vector space. These methods are sometimes also referred to as a Vector Symbolic Architecture (VSA).   Arguably the prototypical VSA is Smolensky's Tensor Product Representation (TPR) [2] but the need for a VSA with constant memory size independent of the complexity of the symbolic structure gave rise to many methods [3] among which HRR is a particularly early and popular one. However, as the authors correctly point out, connectionist models such as the HRR have not been combined with the powerful gradient descent based deep learning of today. It is for this reason that this work is so interesting: it gives compelling evidence and arguments for how HRRs can be employed as a representation inside a modern neural network that is trained by gradient descent on complex data. This comes with clear benefits compared to the TPR-based approaches of recent years published at top conferences (NeurIPS, ICML, ACL; see e.g. [4,5,6]). In addition, the paper is well written with a lengthy analysis.

**Weaknesses**

1.) VSAs with a constant size $d$, such as the HRR, eventually suffer from interferences if the number of bindings exceeds its capacity $m$.  In regular HRRs the capacity increases almost linearly with $d$ [7]. Unfortunately, a discussion on capacity is entirely missing from this work.


**Questions**

1.) What is the vector dimension for the HRR in figure 1? How large does the number of bound terms has to be in this case until the HRR fails? Maybe add this information to the figure description.

2.) $1-\text{cos}(p^* \otimes \hat{s}, c_p)$ is a logical choice but wouldn't it make also sense to use $\text{exp}( \langle p^* \otimes \hat{s}, c_p \rangle)$ (where $\otimes$ is the binding operator)? Modern binary Hopfield Networks (Classic Hopfield Networks can be thought of as TPRs) enjoy an increased capacity using an exponential energy function.

3.) Similarly, $\text{exp}(\langle s, \hat{s} \rangle)$ would be an alternative objective function since it would have a better time complexity. I'd appreciate the authors' comment on this possibility.

4.) After reading, it appears to me that the proposed method can directly be applied to large vocabulary language modelling problems like WikiText-103 (268k vocab) or the Google Billion Word benchmark (793k vocab)? This would be of great practical interest to the NLP research community. Have the authors considered this? The authors may find it useful to add such a possibility to their motivation or conclusion.

**Comments**

typo on line 34, 110, 226, 231

remove empty lines 103, 185

broken sentence on line 84-86

forgotten placeholder on line 295

repetitive lines 88-91 and 221-224 can be removed since it is already stated in 35-37

**References**

[1] Greff, Klaus, Sjoerd van Steenkiste, and Jürgen Schmidhuber. "On the binding problem in artificial neural networks." arXiv preprint arXiv:2012.05208 (2020).

[2] Smolensky, Paul. "Tensor product variable binding and the representation of symbolic structures in connectionist systems." Artificial intelligence 46.1-2 (1990): 159-216.

[3] Schlegel, Kenny, Peer Neubert, and Peter Protzel. "A comparison of vector symbolic architectures." arXiv preprint arXiv:2001.11797 (2020).

[4] I. Schlag, J. Schmidhuber. Learning to Reason with Third Order Tensor Products. Advances in Neural Information Processing Systems (NeurIPS), Montreal, 2018. Preprint: arXiv:1811.12143

[5] Qiuyuan Huang, Paul Smolensky, Xiaodong He, Li Deng, and Dapeng Wu. Tensor product generation networks for deep nlp modeling. In Proceedings of NAACL, 2018. URL https: //arxiv.org/abs/1709.09118.

[6] I. Schlag, K. Irie, J. Schmidhuber. Linear Transformers Are Secretly Fast Weight Programmers. Proc. ICML 2021. Preprint: arXiv:2102.11174.

[7] Plate, Tony A. "Holographic reduced representations." IEEE Transactions on Neural networks 6.3 (1995): 623-641.


---------------------------------------------
Update:

After discussion with the authors, I have decided to increase my score to an 8.

**Time Spent Reviewing:**

6

---

> ### Author Response · Authors · 2021-08-09
> **Reply to sKB5**
>
> First, thank you for the extensive additional related work - particularly on recent TPR based approaches. We agree fully with the reviewer and will add them to our related work and discussion, and the revised manuscript is stronger for it.
>
> We will add reference to the capacity results of Plate and the linear increase with respect to the dimension size d. While Plate’s work did develop theory, we find that the naive HRR does not match Plate’s theory, the number of concepts that can be bound and then retrieved without errors grows sub-linearly for the standard HRR, but when using our complex projection step does exhibit predictable linear behavior once the dimension d is $\geq$ 128 dimensions. We will add results showing the binding capacity more explicitly in the appendix, along with a comparison to the more recent VTB binding showing that VTB and projected HRRs have statistically equal capacity.
>
> Regarding your questions:
>
> 1. Lines 145-146 stated the dimension of 256 used in Figure 1, we will add this to the caption to make it easier to track.
>
> 2. The proposed $\text{exp}( \langle p^* \otimes \hat{s}, c_p \rangle)$ is viable for the $J_p$ equation, we did not explore it though. We had converged onto the cosine similarities scores due to their nature as a scaled dot product that matches the $\hat{s}^\top \mathit{query}^{*}$ behavior of HRRs, but avoids the issues of learning unreasonable large magnitude responses we discussed on lines 197-199.
>
> 3. We did explore $\text{exp}(\langle s, \hat{s} \rangle)$ in early testing, but found it difficult to learn due to the same reasons as $\|s-\hat{s}\|$ (line 188) in that the noise caused by the HRR keeping all items into the fixed dimension space then becomes a contributing factor to the loss, and would lead to degenerate solutions.
>
> 4. We have considered future NLP applications of our work and are actively considering them. We think question answering systems over unstructured text is an excellent example of a problem where our approach may be of great value. A successful HRR approach would seem ideally poised to answer such questions by simply querying the learned sentence vector to retrieve the appropriate object, and by our dot product “cow” example in the paper can also check that the query is not answerable. This could also lead to interesting neural knowledge base hybridization, where we can manually inspect which factual concepts have been retained by the model or manually insert new concepts into the model. This is obviously not trivial, but the possibilities are exciting and we believe it is a good fit to expand upon our work. We will add discussion of future possibilities to the manuscript, thank you for the suggestion.
>
> Thank you for the typo catches and lines, we will fix them! The placehold was for 22 total lines of code.

---

> > ### Comment · Reviewer_sKB5 · 2021-08-11
> > **Reply**
> >
> > Thank you for your response. I'm pleased with answers to 2) and 3). Regarding 4): after looking at the appendix, it has become clear to me that it would not be straightforward to use the HRR for large vocabulary language modelling due to the lack of a calibrated probability. Nevertheless, I fully agree with the authors about the potential of using HRRs to learn more structured neural representations. Lastly, my first question was two-fold and I didn't find an answer yet two the second question.
> >
> > In the context of figure 1: what is the number of binding vectors where an HRR with $d=256$ starts to fail?
> >
> > >While Plate’s work did develop theory, we find that the naive HRR does not match Plate’s theory, the number of concepts that can be bound and then retrieved without errors grows sub-linearly for the standard HRR, but when using our complex projection step does exhibit predictable linear behavior once the dimension d is  128 dimensions.
> >
> > I'd appreciate it if the authors added such experimental results to the appendix and the respective code to their supplemental material.

---

> > > ### Author Response · Authors · 2021-08-11
> > > **Reply^2**
> > >
> > > We are glad our answers to your questions 2 & 3 are satisfying, and appreciate the follow up. Please do let us know if any other questions or this are not fully satisfied.
> > >
> > > >In the context of figure 1: what is the number of binding vectors where an HRR with $d=256$ starts to fail?
> > >
> > > There are multiple ways to look at Figure 1's "failure" point, since it is looking at the average response over all items, so single item failures do not have too large an impact.
> > >
> > > If we look at when do the mean and standard deviation of the responses start to overlap for the present/absent cases, that starts around $n=512$ items. If we look at when the standard deviation starts to approach the other item's mean, that occurs around $n=1,536$. If we look at when the mean response begins to deviate away from the target value of 1/0, that does not start to occur until around $n=2^{16}= 65,536 $! This stability of the average for large $n$ is an important component of our loss component $J_n = \cos{\left(\mathbf{m^{\dagger}} \otimes \hat{\mathbf{s}}, \sum_{\boldsymbol{c}_p \in \mathcal{Y}^p} \boldsymbol{c}_p \right)}$ is implicitly working over an average response of all the negative labels.
> > >
> > > Essentially the distributional average around the desired 1/0 response for present/absent items is _very_ stable, but the tails of the distribution do begin to grow as you try to pack more and more into the single representation.
> > >
> > > We will add this information with additional details into the updated manuscript. We regret that we do not appear to be able to update the supplemental material through OpenReview to give you access to the code immediately. It appears the supplement is also frozen with the PDF while under review.
> > >
> > > >I'd appreciate it if the authors added such experimental results to the appendix and the respective code to their supplemental material.
> > >
> > > We have already added it to our update manuscript. As evidence, please see the below table for error-free recovery of $\geq$97% of all vectors inserted (i.e., after inserting $n$ bounded item pairs, and another $n$ non-inserted distractors, unbinding a valid item is more similar to the original item then any distractor item). The capacity for low $d$ starts at 8 for both Naive HRRs and our improved variant, but our new variant increases $\approx$ linearly up to 384 unique bound pairs, where the original HRR is stuck at 24. Going past $d=8100$ starts to become a memory and compute issue.
> > >
> > > We note the difference between this table and the preceding result is the distinction between how the VSA is used: in the first case we are looking for a consistent average response, in which case small errors are not enough to cause issue, and so large effective capacity is obtained. This below result is for error-free recognition of the original item against $n$ distractors, so the error rate must be considerably less than $\frac{1}{n}$ on a per-item basis in order to pass $n$ total false comparisons.
> > >
> > > | Dimension d | Naive HRR | HRR w/ Proj |
> > > |:-----------:|:---------:|:-----------:|
> > > |          25 |         8 |           8 |
> > > |          64 |         8 |           8 |
> > > |         121 |         8 |          16 |
> > > |         256 |         8 |          16 |
> > > |         484 |         8 |          32 |
> > > |        1024 |        12 |          64 |
> > > |        2025 |        16 |         128 |
> > > |        4096 |        24 |         192 |
> > > |        8100 |        24 |         384 |
> > >
> > > The expanded appendix section on capacity also includes VTB (same high capacity, but required significantly more memory to backpropagate through with current tooling) and the MAP-C VSA (has undesirable zero sub-gradients, and $\approx \frac{1}{2}$ the capacity for error free retrieval).

---

> > > > ### Comment · Reviewer_sKB5 · 2021-08-18
> > > > **Reply^3**
> > > >
> > > > Thank you for your detailed answers and effort to address the issues raised by the reviewers. I think this paper is by far more interesting and better suited than most papers. I also have the impression that the changes due to reviews have improved the paper. I want to see this paper accepted which is why I have increased my score to an 8.

---

> > > > > ### Author Response · Authors · 2021-08-18
> > > > > **Reply^4**
> > > > >
> > > > > Thank you for raising your score! We have been working to integrate all suggestion, and are appreciative of the very thorough and detailed reviews. We do believe they have improved the manuscript, as well as our ideas for future work. Please let us know if there are any other questions or items we can clarify.

---

### Official Review · Reviewer_m8WY · 2021-07-16

**Rating:** 7
**Confidence:** 3

**Summary:**

The authors introduce an improved formulation of learning using holographic reduced representations (HRRs). The paper shows that the improved HRR formulation allows HRRs to be used for eXtreme Multi-Label (XML) classification tasks, and that using an HRR-based approach leads to improved performance over FC, CNN and LSTM baselines. In addition to comparing against baselines, the paper also assess the impact of various hyperparameters.


**Ethical Concerns:**

None.

**Limitations And Societal Impact:**

No societal impact.
As discussed above, I think that the authors did a good job discussing the limitations of intended scope of their contribution.

**Main Review:**

Update:
Thank you for your response. Given the clear responses to all reviewers, and given the discussion of significance with reviewers rafZ and sKB5, I will raise my score to a 7.
--------

Significance: I will start my review by saying that I was previously unfamiliar with holographic reduced representations (HRRs), so it’s difficult for me to judge significance of improving learning in HRRs, and I will defer to the other reviewers as to the value of such a contribution. However, I think this paper does a very good job clearly stating the intended scope (“Our experiments and ablations will focus on the impact of replacing a simple output layer with HRRs and other changes to the HRR approach to better understand its behaviors. State-of-the-art XML performance is not a goal.”), and I believe that these goals are fulfilled by the very thorough experiments. I think the authors do a good job showing improvements in HRR learning, exploring the pros and cons of HRRs, and that this technique could potentially be the basis for future work.


Clarity:  I found section 3 a bit confusing to read. In particular, I don’t think it was a self-contained introduction to HRRs for a reader who is unfamiliar. I would recommend first stating the intended high-level goals of the HRR approach, then showing an example of usage, and then discussing details such as the implementation and properties  (equations 1 and 2).
As a reader, it would be much more helpful to first read about the motivation of HRRs  (disused somewhat starting on line 104), before reading about how they are implemented (lines 93 - 103).


On the other hand, the experiments section (Sec 5) is very clearly written and easy to follow. I appreciate the clarity of this section. All experiments were clearly explained and the experimentation was thorough, comparing to FC, CNN and LSTM baselines (5.1) , examining the impact of hyperparameters (5.3), as well as discussing factors impacting runtime and how they could be improved (5.2). For a paper exploring the performance and behavior of a new technique, I think this work does a good job a) clearly explaining each experiment, b) thoroughly exploring different aspects which contribute to the performance of the new technique, and c) discussing other important considerations (such as runtime)


As I am not very familiar with HRRs, I will have to defer to other reviewers on the significance of the contribution, but I believe the quality of the evaluation is high, the evaluation section of the paper is thorough and clear, and the scope and claims of the work are clearly stated. For these reasons, I would tend to favor acceptance, provided that section 3 was rewritten to be a bit more clear.


Other comments/questions:
- It would be very helpful to show a figure with a concrete example of an XML task, so the reader can understand the task better.

- What other domains/problems might HRRs be suitable for, besides XML? For somebody unfamiliar with HRRs, a discussion of other domains for future work would be very helpful to help give a sense of what future impact might be.



The paper does need a thorough proofread. Below are some minor notes and typos:
- Line 191: what does the following sentence mean? “Because binding many values in a fixed dimension d0  introduces noise, the regression approach attempts to learn this underlying noise that is not meaningful to the actual problem.”
- Table 1: it’s —> its (and other places throughout the paper)
- line 225: to due to  —> in order to ?
- line 110: sybmolic
- line 226: Convoltuional
- line 231: heiarchical
- line 295: XXX lines of code — I assume this is meant to be replaced with an actual number?


**Time Spent Reviewing:**

4

---

> ### Author Response · Authors · 2021-08-09
> **Reply to m8WY**
>
> Thank you for the feedback about the clarity of section three. While we can not fit a full tutorial on HRRs, we will expand section 3 with more details and the appendix with additional diagrams and a worked out example of HRRs to improve the accessibility of the manuscript. This will include a concrete example of the HRR.
>
> We believe HRRs may be imminently suitable for work in scalable Multi-Headed Attention/Transformers (via accelerating the query response), considerable work in natural language processing applications and question/answering systems that require logical reasoning (e.g., subject-noun binding allowing disambiguation, and question-answer systems), and few shot learning (e.g., by representing a class as the addition and binding of HRR concept vectors representing core-concepts vs modifiers of the instantiation). We will add further information about this into the manuscript.
>
> Thank you for the typos, we will fix them and do an additional proofreading of the paper. The XXX should have been 22 lines of code.
>
> >Because binding many values in a fixed dimension d0 introduces noise, the regression approach attempts to learn this underlying noise that is not meaningful to the actual problem
>
> The single feature vector $\hat{\boldsymbol{s}}$ and $\boldsymbol{s}$ are each attempting to represent multiple values in a single space, which necessarily is a lossy operation. As such the exact values stored in $\boldsymbol{s}$ do not necessarily correspond to information about the class labels. However, the regression approach does not capture this, and attempts to minimize the numeric error of all coordinates, even though they may be a function of noise rather than underlying information. The network attempting to learn to predict these noise terms is intrinsically an unhelpful task.
>
> We are glad the reviewer found section 5 easy to read and follow, the positive feedback is greatly appreciated as we endeavor to incorporate the reviewer’s valuable feedback.
>
> We hope the reviewer agrees with the other reviews that our HRR work is significant and addresses a gap in the literature.

---

### Official Review · Reviewer_LW6b · 2021-07-16

**Rating:** 6
**Confidence:** 4

**Summary:**

The paper makes an incremental change to an early cognitive model of symbol binding which positions the work in the context of other end-to-end neurosymbolic approaches that make use of embeddings.

**Main Review:**

Without further evaluation either theoretical or through comparative experimental evaluations of the benefits of the proposed increment wrt the related work in this area, I don't believe that the paper can be accepted.
Having double checked the supplementary materials and the experimental results, I now appreciate the potential of the approach despite the lack of comparative evaluation results. I have increased my score accordingly.

**Time Spent Reviewing:**

3

---

> ### Author Response · Authors · 2021-08-09
> **Reply to LW6b**
>
> We do not understand the reviewer’s position. We have discussed the math about why we chose to make a projection step through lines 122-130, and all of section four gives thorough mathematical explanations for why and how the loss function design enables leveraging the symbolic properties of the HRR while avoiding potential learning problemes through gradient based training (e.g., not using the L2 loss and using cosine). We feel our experiments are quite thorough, covering three different types of network architectures, over 10 datasets, and ablating over 4 different design choices. Not to mention the cases which are included in words because the experiments resulted in random-guessing performance (e.g., training without the projection) that are further results. Our related work has established that there is little work in this space, and other reviewers have provided important related work, but does still confirm that this is a largely unexplored area that deserves more attention from the ML community.

---

> > ### Comment · Reviewer_LW6b · 2021-08-18
> > **Incremental contribution and limited evaluation and not accessible to wider audience**
> >
> > In my view the contribution of the paper is incremental and rather straightforward (Section 3.1). Since the motivation for the paper is neurosymbolic AI, I'd have expected comparisons to be drawn with e.g.:
> > Tensor Product Representations,
> > Neuro-Symbolic Concept Learner,
> > Logic Tensor Networks,
> > or other differentiable approaches combining embedding learning with symbolic knowledge, including work on autoencoders using disentanglement.
> > Some of the above have already been combined with powerful gradient descent based deep learning of today: https://arxiv.org/abs/2012.13635.
> > Instead, the paper's empirical evaluation is restricted to traditional neural models which are then extended with HRR alone.
> > So, it is incremental with a limited evaluation and I don't think that it is particularly well written; it is not accessible to someone who isn't familiar with the specific HRR approach.

---

> > > ### Author Response · Authors · 2021-08-18
> > > **Rebuttal**
> > >
> > > We will be sure to include the Neuro-Symbolic Concept Learner (NSCL) and Logic Tensor Networks into our related work, we do however feel that these works differ significantly from our own. Both are exploring differentiable first-order logic representations. Such work is clearly valuable and capable, but we believe is also more involved than our work in technique. Thank you for drawing them to our attention and the valuable framing and context they will provide the reader.
> > >
> > > We do hope the reviewer will consider our contributions from both section 3.1 and section 4, which we feel are together non-trivial, significant, and unique from other works. Our belief is also that simple changes with meaningful improvement, as our projection step shows, are more valuable in distilling what is necessary to its minimal components. In discussion with other reviewers we have already included TPRs into the revision, and discussion about how they are related, but the expanding dimension size does put them into a different category in terms of utility, and that a fixed-sized representation is valuable. We have also added exposition and results on binding capacity of our improved HRR compared to other VSA and their favorable paired binding capacity and properties, as shown in reply to sKB5.
> > >
> > > It is always difficult to include details in a limited space to maximize accessibility to readers, especially when working with not widely known methods like HRRs. Thanks to reviewers’ detailed feedback we have replied to rafZ with additional appendix context to introduce the reader to why and how HRRs work.
> > >
> > > We respectfully disagree that our evaluation is limited, we argue still that it is quite thorough over many datasets, architecture types, and ablations. Given HHRs already established utility in purely symbolic AI tasks, we do feel it appropriate to explore HRRs for tasks that are not intrinsically symbolic - which is congruent with the captioning, classification, and clustering results of the NSCL and LTN works you provided.

---

> > > > ### Comment · Reviewer_LW6b · 2021-08-23
> > > > **High potential value**
> > > >
> > > > I appreciate the authors' competent reply and the enthusiasm of the other reviewers about the potential value of this work and I have adjusted my score accordingly having spent more time checking the supplementary materials and experimental results.

---

> > > > > ### Author Response · Authors · 2021-08-24
> > > > > **Thank you**
> > > > >
> > > > > Thank you for raising your score, we are glad we were able to satisfy your concerns in discussion and the extra content of our appendix. We appreciate the time spent in reading our paper and the review discussions.

---

### Official Review · Reviewer_rafZ · 2021-07-19

**Rating:** 7
**Confidence:** 4

**Summary:**

This paper investigates the technique of “Holographic Reduced Representation” (HRR), developed in the early 90’s by Tony Plate as a tool for symbolic reasoning, for use in modern neural network architectures trained using backpropagation. The authors find that Plate’s original formulation leads to low accuracy in this setting and present a simple fix that improves performance. The authors apply the technique to classification problems in which the outcome is a high-dimensional binary vector and show that HRRs can be used effectively in conjunction with gradient based training methods in this setting. The paper is well written, technically sound, and discusses an interesting avenue for bridging classic work on connectionist AI with more modern techniques.

**Ethical Concerns:**

As noted above, I see no particular ethical concerns beyond those present in any AI research.

**Limitations And Societal Impact:**

This work is fairly agnostic with respect to applications and beyond the possible societal impacts that are inherent in any work about developing better classification algorithms I see no particular causes for concern about this work. I do not think any further action on the part of the authors is required.

**Main Review:**

[Summary and Contribution] The 90’s saw significant interest in “symbolic” approaches to AI. The basic idea of these techniques is to represent symbols as vectors and then to define operators that allow one to encode various relations between symbols. The primary benefit of such approaches, which are collectively known as “vector symbolic architectures” (VSA), is that they provide a structured way to encode complex relationships between objects into a single high-dimensional point. Tony Plate’s “Holographic Reduced Representation” (HRR) is one particular approach to this problem. These techniques subsequently fell out of vogue in the AI/ML community and were largely supplanted by neural networks trained using backpropagation. The primary contribution is a simple modification to the original approach of Plate that improves the storage capacity of HRR and leads to better accuracy when training in a neural network with backprop. The authors also present an interesting application of the technique to “extreme multi-label classification,” in which HRR essentially functions as a form of label compression, and show that it leads to results that are at least in the same ballpark as state-of-the-art techniques. While the contribution is fairly simple, this is an interesting piece of work connecting two different areas of AI and would likely be of interest to a fairly diverse audience of readers. The work additionally contains some useful practical guidance on that cannot be found elsewhere in the literature.

[Originality] It is worth mentioning in the related work section that Plate himself discusses using HRRs with gradient based systems in his book “Holographic Reduced Representation” (see ch. 5 “Using Convolution Based Storage in Systems that Learn”). However, this work considers incorporating HRRs with additional types of network architecture and provides new insight into the choice of cost function. Additionally, there has been substantial work on using similar “connectionist” architectures in learning systems which warrants some brief discussion. See for example: Imani et. al “VoiceHD: Hyperdimensional computing for efficient speech recognition,” Kleyko et. al “Classification and recall with binary hyperdimensional computing: Tradeoffs in choice of density and mapping characteristics.” The work by Imani et. al also considers a combination of VSA representation and trainable neural network albeit in a very different way than is discussed here.
[Quality/Clarity] This work is sound technically and is generally well written. I have the following comments:
•	In general, the paper would be stronger if there was more formalism about what kinds of properties we need the HRRs to satisfy and some analysis of why they meet those requirements. In a nutshell, we need to be able to query the HRR to ask if a certain relation is present – it would be good to provide some brief description of the mathematics behind why HRRs are able to support this. For instance, on line 107, why should we expect that <S,cat> ~ 1 but <S,cow> ~ 0? A brief sketch of why this works out mathematically would be fine.
•	Related to the above, it would be nice to provide some analysis of why the proposed modification improves query accuracy. There are several works that provide analysis of similar techniques and could be adapted to this setting. See for example, Frady et. al “A Theory of Sequence Indexing and Working Memory in Recurrent Neural Networks” or Thomas et. al “Theoretical Foundations of Hyperdimensional Computing” who both provide detailed analyses of related schemes.
•	The choice of d (dimension of the HRR) seems fairly arbitrary here. The relationship between d, the number of relations to encode, and the probability of correctly answering a query has been characterized for several other related methods (see either of the two works referenced above). It would be good to provide some more formal guidance on how one should choose this parameter.
•	HRR is one particular instance of a broader family of methods for symbolic reasoning on vector representations – this line of work is often goes by the name “vector symbolic architectures” or “hyperdimensional computing.” Moreover, there are VSA architectures for which the “binding” operator (circular convolution here) is simpler computationally and has an exact inverse. I would have liked to see a bit of discussion as to why HRR might be preferred over other VSA architectures. A detailed discussion is probably beyond the scope here but it would be nice to at least discuss this question a bit and point the reader to some of the other literature in this field.
•	Minor comments/Typos:
o	The requirement that the embeddings be sampled from a N(0,1/d) distribution is sufficient, but not necessary.
o	Line 85 “is high architecture” – line 94 “it’s” – line 154 “response tightly” – line 225 “HRR to due to show” – line 295 “implemented with XXX lines of code – line 316 “the largeness of” – line 341 “is queries against”
[Significance] This work represents an important first step in bringing classic work on connectionist/symbolic AI into the fold of modern machine learning. The connectionist approach has a lot of interesting qualities and has recently seen a resurgence of interest from the hardware community, who have made extensive use of such techniques to offer low-power alternatives to conventional ML algorithms like SVMs and MLPs. However, these techniques are under-studied by the main-stream ML community and their utility for solving complex learning tasks remains unclear. This work provides novel and useful practical guidance and will likely spur greater interest in this area by other ML researchers. From that perspective, the authors may want to provide a bit more discussion as to why HRR/VSAs are a promising direction in general and warrant further attention. What do these techniques offer that conventional approaches based on deep neural networks alone do not? On the other hand, do these methods have clear limitations? What do DNNs offer HRRs? Are there reasons to prefer HRRs over other methods for label compression? These questions are all speculative, but offering some opinions on them would generate good material for future discussion.


**Time Spent Reviewing:**

3

---

> ### Author Response · Authors · 2021-08-09
> **Reply to rafZ**
>
> Thank you for the  Imani et. al and Kleyko et. works, which indeed are related in their use of VSAs and we will add them to our related works.
>
> We will add a worked example of how and why HRRs work to the appendix to make the work more accessible, thank you for the suggestion.
>
> We believe our analysis that the complex unit magnitude projection allows the use of the pseudo-inverse operation to equal to true inverse, while avoiding numerical instability, helps to explain why accuracy is improved. We will endeavor to expand on this consulting the Frady et. al & Thomas et. al references, thank you for bringing them to our attention.
>
> The choice of d was fairly arbitrary, and our ablations (e.g., Figure 4) show a strategy that we would recommend for the context of learning with HRRs: start with a dimension of 512, and double the dimension size if performance is not sufficient. The original paper by Plate has a theoretical analysis of capacity that we will reference in an updated version of the paper. We have conducted new experiments that show naive HRRs capacity does not improve linearly in the case of $\sum_{i=0}^k \operatorname{bind}(\boldsymbol{x}_i, \boldsymbol{y}_i)$  until very large dimension sizes $\geq 2048$ where our approach begins to increase linearly in capacity (matching plates theory) at $d \geq 128$. We will add a section to the appendix discussing the capacity of HRRs before/after our projection step.
> NOTE: We have slightly edited the above paragraph, it had some original draft language that was not corrected when submitted. We apologize for any inconvenience this may have caused. The draft was written before we had conducted the new experiment and realized that naive HRRs do not math Plate's theoretical expectation for linear improvement in capacity, at least not until very large values of $d$, and operating at 32-bit float precision.
>
> We will add a reference to Schlegel, K., Neubert, P., & Protzel, P. (2020). A comparison of Vector Symbolic Architectures.  To help with a discussion about other VSAs, and the following reasons we went with HRRs compared to other options: We need fixed-length representations that are composed of differentiable operations over real-valued vectors, which leaves only VTB, HRRs, and MAP-C as potential options. The MAP-C approach has low efficacy as it uses element wise multiplication (as shown in Schlegel). Initial VTB experiments required significantly more memory without writing custom kernels for the binding operation. This left only HRR as the remaining option, which has the benefit of being widely used in cognitive science to provide additional evidence of HRR’s utility.
>
> Thank you for all typo catches! We agree with all of them and they will be corrected.
>
> > bit more discussion as to why HRR/VSAs are a promising direction in general and warrant further attention
>
> Thank you for the suggestion, we will incorporate into the appendix a longer section that discusses these in greater detail, we agree with the reviewer it will help better frame the value of our work to readers. We will also add a paragraph to the main text with a short version of this discussion to reference the appendix for further details. We note in particular that that:
> 1. Our appendix section Inference for XML with HRR did discuss the practicality of this specific task in future work, as it allows recasting the problem of classification as a Maximum Inner Product Search (MIPS) that has many efficient algorithms (e.g., HNSW, IVFQ).
> 2. HRRs allow for unique kinds of adaptive parallelism, where the user/code may decide how many classes to check for from the prediction $\hat{\boldsymbol{s}}$ at runtime, allowing for situational reduced cost and easy distribution of prediction effort.
> 3. Related to your own suggestions, we will make note of existing VSA work leveraging them for hardware applications by  Imani et. al and Kleyko et. al as well as the valuable suggestions on TPR augmented models by reviewer sKB5.

---

> ### Author Response · Authors · 2021-08-16
> **Worked Example of why HRRs work**
>
> We would like to provide the reviewer with a partial excerpt of the worked example of HRRs we have added to the appendix. The beginning section includes diagrams we can not enter into OpenReview, but it provides an example of how one may think of an object (i.e., a dog) represented by symbolic hierarchies (Legs, body, and head, with the head made of a mouth, noes, and eyes) to illustrate the high-level abilities of HRRs. The below excerpt is then used to demonstrate that math was to why such manipulations are possible. We again thank the reviewer for the suggestion.
>
> As an example we will reproduce the excellent illustrative worked example by \citet{Plate1995}, followed by a new derivation showing the same nature when distractor terms are included in the statement.
>
> Consider a $d=3$ dimensional space, where we wish to compute $\boldsymbol{c}^\dagger \otimes (\boldsymbol{c} \otimes \boldsymbol{x})$, we will get the result that:
>
>
> \begin{equation}
> \boldsymbol{c}^\dagger \otimes (\boldsymbol{c} \otimes \boldsymbol{x}) =
>     \begin{bmatrix}
>     \begin{array}{r}x_{0}\left(c_{0}^{2}+c_{1}^{2}+c_{2}^{2}\right)+x_{1} c_{0} c_{2}+x_{2} c_{0} c_{1}+x_{1} c_{0} c_{1} \\ +x_{2} c_{1} c_{2}+x_{1} c_{1} c_{2}+x_{2} c_{0} c_{2}\end{array} \\\\
>     \begin{array}{r}x_{1}\left(c_{0}^{2}+c_{1}^{2}+c_{2}^{2}\right)+x_{0} c_{0} c_{1}+x_{2} c_{0} c_{2}+x_{0} c_{0} c_{2} \\ +x_{2} c_{1} c_{2}+x_{0} c_{1} c_{2}+x_{2} c_{0} c_{1}\end{array} \\\\
>     \begin{array}{r}x_{2}\left(c_{0}^{2}+c_{1}^{2}+c_{2}^{2}\right)+x_{0} c_{0} c_{1}+x_{1} c_{1} c_{2}+x_{0} c_{1} c_{2} \\ +x_{1} c_{0} c_{2}+x_{0} c_{0} c_{2}+x_{1} c_{0} c_{1}\end{array}
>     \end{bmatrix}
> \end{equation}
>
> There are two simplifications that can be done to this resulting matrix by exploiting the fact that all elements in the matrix are sampled according to the distribution $x \sim \mathcal{N}(0, \frac{1}{d})$. First there is the pattern $x_i \cdot \prod_{i=0}^{d-1} c_i^2$ on the left hand side. The sum of squared normals will in expectation be equal to 1, but we will subtract that value in our change of variable to create $\xi=\left(c_{0}^{2}+c_{1}^{2}+\cdots+c_{d}^{2}\right)-1$, which will then have the distribution $\xi \sim N\left(0, \frac{2}{n}\right)$. Second, the right-hand side will have $d (d-1)$ products of the form $x_i c_j c_k, \forall j \neq k$. Summing all of these into a new variable $\eta_i$ products $\eta_{i}  \sim \mathcal{N}\left(0, \frac{d-1}{d^2}\right)$. Inserting $\xi_i$ and $\eta_i$ we get:
>
> \begin{equation}
> \boldsymbol{c}^\dagger \otimes (\boldsymbol{c} \otimes \boldsymbol{x}) =
> \left[\begin{array}{l}x_{0}(1+\xi)+\eta_{0} \\\\ x_{1}(1+\xi)+\eta_{1} \\\\ x_{2}(1+\xi)+\eta_{2}\end{array}\right]=(1+\boldsymbol{\xi}) \tilde{\mathbf{x}}+\tilde{\boldsymbol{\eta}}
> \end{equation}
>
> Since both $\xi_i$ and $\eta_i$ have a mean of zero, we get the final result that $\mathbb{E}[\boldsymbol{c}^\dagger \otimes (\boldsymbol{c} \otimes \boldsymbol{x})] = \boldsymbol{x}$, allowing us to recover a noisy approximation of the original bound value. The communicative and associative properties of the HRR's construction then extend this to the more complex statements that are possible, and accumulate the noise of the resulting variables.
>
>
> To demonstrate this, we will perform another example with $\boldsymbol{c}^\dagger \otimes (\boldsymbol{c} \otimes \boldsymbol{x} + \boldsymbol{a} \otimes \boldsymbol{b})$. This will be performed with $d=2$ in order to avoid visual cluster, and results in the equation:
>
> \begin{equation}
> \begin{array}{lcc}
> \boldsymbol{c}^\dagger \otimes (\boldsymbol{c} \otimes \boldsymbol{x} + \boldsymbol{a} \otimes \boldsymbol{b}) & = &
>     \left[\begin{matrix}\frac{c_{0} \left(a_{0} b_{0} + a_{1} b_{1} + c_{0} x_{0} + c_{1} x_{1}\right) - c_{1} \left(a_{0} b_{1} + a_{1} b_{0} + c_{0} x_{1} + c_{1} x_{0}\right)}{\left(c_{0} - c_{1}\right) \left(c_{0} + c_{1}\right)}\\\\ \frac{c_{0} \left(a_{0} b_{1} + a_{1} b_{0} + c_{0} x_{1} + c_{1} x_{0}\right) - c_{1} \left(a_{0} b_{0} + a_{1} b_{1} + c_{0} x_{0} + c_{1} x_{1}\right)}{\left(c_{0} - c_{1}\right) \left(c_{0} + c_{1}\right)}\end{matrix}\right] \\\\
>     & = &
>     \left[\begin{matrix}\frac{\color{red}{a_{0} b_{0} c_{0} - a_{0} b_{1} c_{1} - a_{1} b_{0} c_{1} + a_{1} b_{1} c_{0}} \color{black}{+ c_{0}^{2} x_{0} - c_{1}^{2} x_{0}}}{c_{0}^{2} - c_{1}^{2}} \\\\ \frac{\color{red}{- a_{0} b_{0} c_{1} + a_{0} b_{1} c_{0} + a_{1} b_{0} c_{0} - a_{1} b_{1} c_{1}}\color{black}{ + c_{0}^{2} x_{1} - c_{1}^{2} x_{1}}}{c_{0}^{2} - c_{1}^{2}}\end{matrix}\right]
> \end{array}
> \end{equation}
>
> Notice that the red highlighted portion of the equation is the product of independent random variables, meaning two important properties will apply: $\mathbb{E}[X Y] = \mathbb{E}[X] \cdot \mathbb{E}[Y]$ and $\operatorname{Var}(X Y)=\left(\sigma_{X}^{2}+\mu_{X}^{2}\right)\left(\sigma_{Y}^{2}+\mu_{Y}^{2}\right)-\mu_{X}^{2} \mu_{Y}^{2}$. Because these random variables have a mean $\mu = 0$, the products result in a new random variable with the same mean and reduced variance as the original independent components. The first property gives
>
> $$\mathbb{E}[X Y] = \mathbb{E}[X] \cdot \mathbb{E}[Y] = 0 \cdot 0 = 0$$
>
> and the second property gives:
>
> $$\operatorname{Var}(X Y)=\left(\sigma_{X}^{2}+\mu_{X}^{2}\right)\left(\sigma_{Y}^{2}+\mu_{Y}^{2}\right)-\mu_{X}^{2} \mu_{Y}^{2} = \left(\frac{1}{d}+0\right)^2 \left(\frac{1}{d}+0\right)^2-0 = \frac{1}{d^4}$$
>
> That reduces each product of $a_i b_j c_k$ into a new random variable with a mean of zero, and then the sum of these random variables, due the the linearity of expectation, will be a new random variable with an expected value of zero. So in expectation, the highlighted red terms will not be present (but their variance due to noise will cause errors, though the variance is harder to quantify due to reuse of random variate across the products). Thus we get the expected result of:
>
>
> \begin{equation}
>     \left[\begin{matrix}\frac{c_{0}^{2} x_{0} - c_{1}^{2} x_{0}}{c_{0}^{2} - c_{1}^{2}}\\\\ \frac{c_{0}^{2} x_{1} - c_{1}^{2} x_{1}}{c_{0}^{2} - c_{1}^{2}}\end{matrix}\right] = \left[\begin{matrix}x_0 \\\\ x_1\end{matrix}\right]
> \end{equation}
>
> Which recovers the original $\boldsymbol{x}$ value that was bound with $\boldsymbol{c}$, even though additional terms (e.g., $\boldsymbol{a}\otimes \boldsymbol{b}$ are present in the summation).

---

### Decision · Program_Chairs · 2021-09-28

**Decision:**

Accept (Spotlight)

**Comment:**

This paper revisits the use of Holographic Reduced Representation (HRR) as a way to combine symbolic reasoning with a neural system. It proposes a new method to efficiently train HRRs. The authors demonstrate that replacing a traditional output layer of a network with a semantically meaningful HRR layer is very effective, in terms of drastically reducing the output layer size and also reducing the overall network size. The results suggest that this is a promising approach that will likely spark future work.

The paper led to substantial discussion among the reviewers and authors, and the authors' informative clarifications helped make all reviewers much more confident in the accuracy and value of the work.

With the authors incorporating their insightful replies into the final version of the paper, this will make a solid NeurIPS paper with fairly broad interest.

**Consistency Experiment:**

NeurIPS has a long history of experimentation. In 2014, NeurIPS ran an experiment in which 10% of submissions were reviewed by two independent committees to quantify the randomness in the review process. This year, we repeated a variant of this experiment to see how the quality of the review process has changed over time.  This paper was part of the experiment and was therefore assigned to two committees (consisting of reviewers, an Area Chair, and a Senior Area Chair) that reached independent decisions.  If both committees made the same recommendation, this recommendation was followed. If a single committee recommended acceptance, the paper was accepted (with the exception of a few cases in which the other committee identified what we considered a fatal flaw, e.g., an error in a key result).

This copy’s committee reached the following decision: **Accept (Spotlight)**

The other committee assigned to the paper recommended **Accept (Poster)**.  You can find the other set of reviews, along with any follow up discussion with the authors here:
https://openreview.net/forum?id=zcrC_XDUFd